# Genomic diversity landscape of the honey bee gut microbiota

Kirsten M. Ellegaard [1] & Philipp Engel [1]

The structure and distribution of genomic diversity in natural microbial communities is largely unexplored. Here, we used shotgun metagenomics to assess the diversity of the honey bee gut microbiota, a community consisting of few bacterial phylotypes. Our results show that most phylotypes are composed of sequence-discrete populations, which co-exist in individual bees and show age-specific abundance profiles. In contrast, strains present within these sequence-discrete populations were found to segregate into individual bees. Consequently, despite a conserved phylotype composition, each honey bee harbors a distinct community at the functional level. While ecological differentiation seems to facilitate coexistence at higher taxonomic levels, our findings suggest that, at the level of strains, priority effects during community assembly result in individualized profiles, despite the social lifestyle of the host. Our study underscores the need to move beyond phylotype-level characterizations to understand the function of this community, and illustrates its potential for strain-level analysis.

[1] Department of Fundamental Microbiology, University of Lausanne, 1015 Lausanne, Switzerland. Correspondence and requests for materials should be addressed to K.M.E. (email: kirsten.ellegaard@unil.ch) or to P.E. (email: philipp.engel@unil.ch)

 1

Most bacteria live in genetically diverse and highly complex communities under natural conditions, and their evolution and ecology is dictated by both external environmental factors and internal microbial interactions. However, surprisingly little is known about how diversity is structured and spatially distributed in natural microbial communities. Are bacteria organized into genetically and ecologically congruent units, akin to the species of the eukaryotic world? And if so, how can we delineate these units? Do closely related lineages co-exist spatially, and if so, how? Addressing these questions is fundamental for understanding the evolution and function of natural bacterial communities[1].

16S rRNA amplicon sequencing is the most commonly used culture-independent method to analyze bacterial communities and has provided valuable insights into the distribution of bacteria across habitats[2]. However, for studying bacterial evolution and function, the 16S rRNA gene lacks resolution. First of all, the same divergence in the 16S rRNA sequence can correspond to vastly different levels of relatedness at the genomic level, even within the same bacterial species[3,4]. Secondly, adaptation of bacterial populations to specific environmental conditions is often linked to functions in the accessory gene pool, which cannot be inferred from amplicon sequencing data[1]. Shotgun metagenomics is therefore becoming an increasingly popular method because it provides both strain-level taxonomic resolution and information about the functional repertoire of the sequenced community[5,6].

Despite the on-going debate on the existence and delineation of bacterial species, read recruitment of shotgun metagenomic data to reference genomes[7] has provided evidence that many bacterial populations are organized into discrete genetic clusters[8]. These so-called 'sequence-discrete populations' (SDPs) are defined by a genetic discontinuity in relatedness between a population of strains and the rest of the community[3,7–10]. Evidence for the broad existence of SDPs in bacteria was recently obtained based on a pairwise average nucleotide identity (ANI) analysis on more than 90,000 sequenced genomes from the NCBI genome database[11]. However, in natural populations, SDP analyses have so far only been conducted in a small number of studies, and almost exclusively from aquatic habitats[3,7–10], leaving the question somewhat open as to how general this pattern is. Since SDP analysis requires representative reference genomes and high sequencing coverage at the taxon level, it is in practice often limited to a very small fraction of the members of a given community.

The successful analyses of shotgun metagenomic data, including de novo assembly, variant calling, and SDP analysis, depends largely on sequencing depth and community complexity[5]. Therefore, the honey bee gut microbiota has great potential as a model for studying natural bacterial populations due to the remarkably simple and conserved composition of the community[12,13]. Multiple studies employing 16S rRNA amplicon sequencing have confirmed that the honey bee gut microbiota is composed of only 8–10 phylotypes (i.e. clusters of 16S rRNA sequences with ≥97% sequence identity), belonging to the Firmicutes, Actinobacteria and Proteobacteria phyla, which typically make up >95% of all 16S rRNA sequences[14–16]. These phylotypes appear to be specific to the honey bee gut and dominate the community regardless of the geographic origin of the sampled bees, their age or the season[15–18]. However, genome sequencing has revealed the presence of highly divergent strains and extensive gene content diversity within several phylotypes, indicating that the community is considerably more complex than 16S rRNA amplicon sequencing would suggest[19–21]. From a taxonomic perspective, the extensive intra-phylotype diversity has resulted in some phylotypes now harboring multiple named species, and others harboring highly divergent lineages with the same species name[22]. Naming of species within phylotypes has so far mostly been motivated by divergence in the non-ribosomal part of the genomes and the aforementioned differences in metabolic profiles[23], but also on ANI[24]. However, the natural population structure within phylotypes has not been systematically investigated, and it is unclear to what extent the named species represent discrete evolutionary lineages (SDPs). Moreover, metabolic profiling has only been done on a small number of isolates[19,23,25], raising the question whether such metabolic features do in fact represent species core repertoire. Regardless, there is increasing evidence that the intra-phylotype diversity is of functional relevance for the bee gut microbiota. For example, divergent strains of the phylotype corresponding to *Gilliamella* exhibit different capacities to degrade pectin, a major plant glycan of pollen[21]. Similarly, it has been shown that strains of one of the two *Lactobacillus* phylotypes (originally named 'Firm5') differ in their ability to metabolize sugars commonly found in the honey bee diet[23]. How these strains distribute in the host population has remained elusive, but may influence the interaction of the bee gut microbiota with its host[26,27], with possible effects on the health status of this important pollinator species.

In the current study, we used shotgun metagenomic sequencing to comprehensively analyze the genomic diversity landscape of the honey bee gut microbiota, at three taxonomic levels: phylotype, SDP and strain (Fig. 1a). By generating metagenomic samples from individual honey bees, we analyzed the distribution of genomic diversity within and between colonies (Supplementary Fig. 1). Our analysis revealed remarkably high levels of genomic diversity within colonies, of which only a fraction was carried by individual bees, resulting in marked functional differences, and highlighting the need to move beyond phylotype-level analysis in future studies.

## Results

**Extensive metagenomic read recruitment to a reference database.** We shotgun sequenced the gut microbiota of 54 bees of three different age groups, sampled from two adjacent colonies, one of which was sampled over two consecutive years (Fig. 1b). To this end, we established a DNA extraction protocol that allowed for the enrichment of bacterial cells in samples originating from single homogenized bee guts (hindgut), and constructed a reference genome database containing genomic data for all major phylotypes of the honey bee gut microbiota (Fig. 1c, Supplementary Data 1). The enrichment protocol efficiently reduced host-derived DNA in most samples, with an average of 13% of the reads mapping to the honey bee genome (Supplementary Fig. 2a). Moreover, ~90% of the remaining reads mapped to the reference genome database in the majority of samples, regardless of honey bee age, indicating that the current database is representative of most of the community (Supplementary Fig. 2b). One sample (DrY2_W5) did contain an abnormal amount of unmapped non-host reads (71%), and was therefore removed from all downstream analyses related to the core microbiota composition and distribution.

To identify bacterial lineages not represented in the current database, orthologs of 33 universally conserved gene families[28] were extracted from de novo assemblies of non-host reads, and the subset of these sequences without a close hit to the current genomic database (<95% nucleotide sequence identity) was further analyzed. The median number of such orthologs per gene family ranged from 0–3 per sample, indicating that relatively few additional taxa were present with sufficient coverage for assembly in a given sample (Supplementary Data 2). A blast search against the NCBI non-redundant database showed that the taxa not represented in the current database belong to three

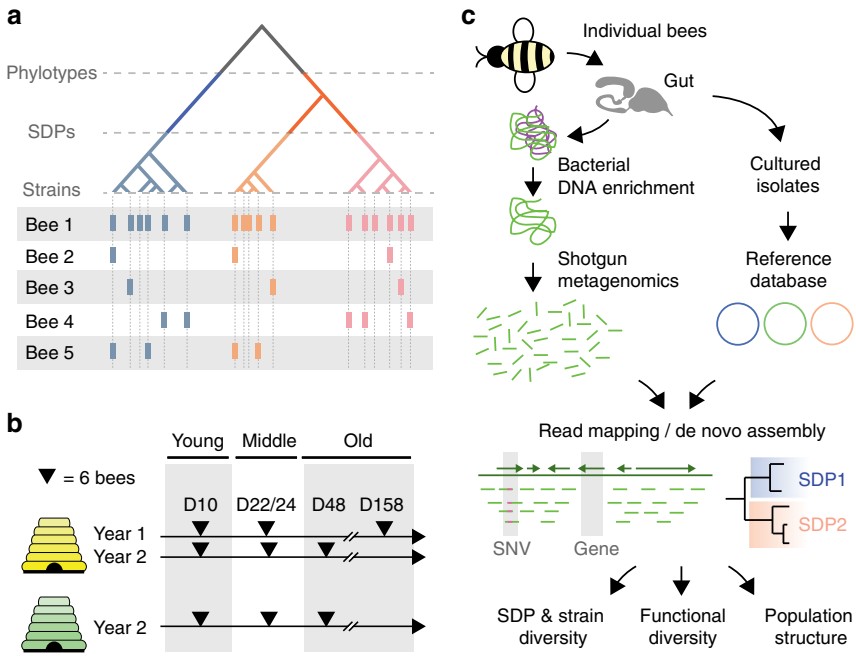

**Fig. 1** Overview of the experimental design and approach. **a** Three levels of diversity were explored in the current study: (i) phylotypes, i.e., bacteria with about 97% similarity in the 16S rRNA gene (as defined by 16S rRNA amplicon sequencing studies), (ii) SDPs (sequence-discrete populations), i.e. divergent sub-lineages contained within phylotypes, (iii) strains, i.e. any genomic diversity occurring within SDPs. A hypothetical example of possible distributions of strains across bees is shown with colored bars. **b** In total, 54 individual honey bees of different age and from two different colonies were sampled for shotgun metagenomics analysis. At each time point, six bees per colony were sampled. Colony 1 was sampled in two consecutive years. Bees sampled 10 days, 22/24 days, and 48/158 days after emergence are considered as 'young', 'middle-aged', and 'old', respectively. The 'old' bees correspond to winter bees as the sampling started in late September. **c** Guts of individual bees were dissected and subjected to a customized protocol enriching for bacterial DNA, which was then shotgun sequenced using Illumina. In parallel, a reference database of high-quality draft and complete genomes of bee gut bacterial isolates was established. The downstream analysis of the metagenomic data was based on both mapping against the reference database and de novo assembly. An overview flowchart of the entire analysis illustrating how genomic and metagenomic data was integrated is given in Supplementary Fig. 1

different classes of Proteobacteria (alpha, beta and gamma). Hits to *Bartonella*, *Commensalibacter* and *Snodgrassella* were found in several samples, indicating that these bee gut microbiota phylotypes still lack some representation in the current database. In addition, hits to phylotypes not considered as part of the honey bee gut microbiota occurred sporadically (e.g. *Sphingomonas* sp, *Acinetobacter* sp., and *Serratia* sp.) (Supplementary Data 2).

Altogether, these analyses show that bacteria-enriched metagenomes can be obtained from individual honey bees, and that a large proportion of the diversity in these metagenomes is represented in our reference database, justifying its utilization for downstream analyses.

**Core members are composed of sequence-discrete populations.** To begin our characterization of the genomic diversity landscape of the honey bee gut microbiota, we first investigated whether phylotypes of this community harbor SDPs and hence are organized into genetically congruent units. To this end, candidate SDPs were identified in the reference database by clustering genomes based on phylogenetic relationships and pairwise average nucleotide identities (ANI) (see methods and Supplementary Fig. 3). With this approach, we identified candidate SDPs for four of the five core phylotypes and for *Bartonella apis*, whereas no candidate SDPs were found for the core phylotype *Snodgrassella alvi* (Fig. 2, Supplementary Figs. 4 and 5, see methods). Since SDPs inferred from sequenced isolates could potentially result from biases in database representation, we validated the candidate SDPs using the metagenomic data (see methods and Supplementary Fig. 3). Briefly, orthologs of core gene families were extracted from the de novo assemblies of the

metagenomic samples and aligned to the core sequences from the database. For true SDPs, the metagenomic core gene orthologs are expected to have a much higher similarity to one particular SDP relative to any other SDP (resulting in a 'gap-zone' as described in[8]). Indeed, we found a clear separation in the distribution of sequence similarity between the highest and second-highest scoring SDP for four of the five phylotypes with candidate SDPs (Fig. 2). Only for *B. apis*, the distributions overlapped (Fig. 2f). In this case, the wider curve for the highest-scoring SDP also shows that the current database does not fully grasp the diversity present in the metagenomic samples, as corroborated by the presence of *B. apis*-like bacteria in the unmapped reads (Supplementary Data 2). Thus, it is possible that SDPs are present in *B. apis*, although with much less separation compared to the other phylotypes. A considerable fraction of the bifidobacteria in the metagenomic samples also lacked a close relative in the database (Fig. 2c). However, the two SDPs (Bifido-1 and Bifido-2) were still discrete due to the large divergence between them.

Since all samples in the current study were derived from the same geographic location, we additionally re-analyzed a previously published metagenomic dataset, which consisted of pooled honey bee guts sampled from a colony in the Unites States[21]. Strikingly, as for the Swiss samples, most of the genomic diversity in this sample was represented by our reference database, with only 9% of the reads unmapped. Despite a general shift in abundance towards Proteobacteria in this sample, all SDPs identified in the current study were found to be present and discrete (Supplementary Fig. 6b–d). As for the metagenomes from Switzerland, most orthologs of the 33 universally conserved gene families[28] without a close hit to the database still had best

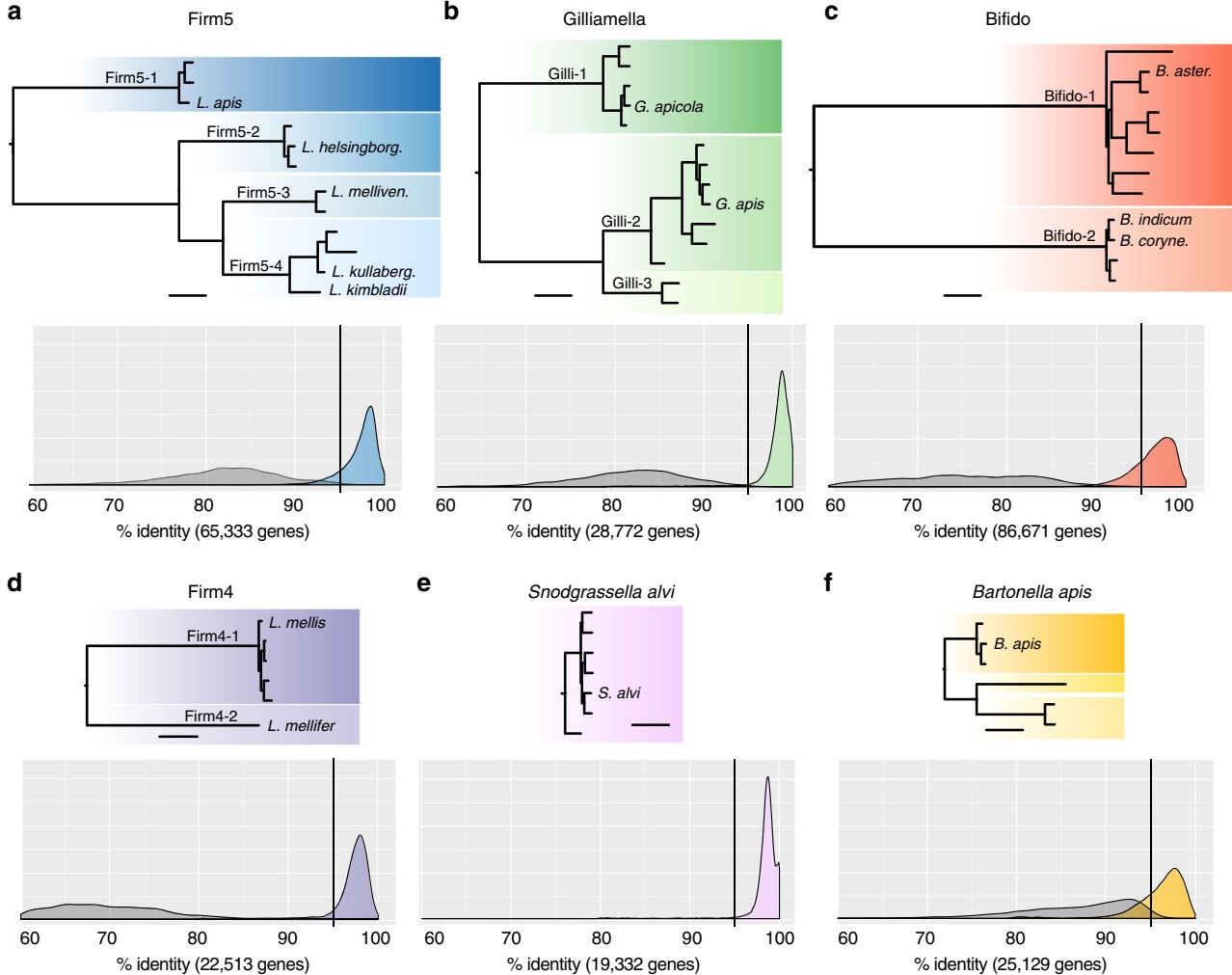

**Fig. 2** Core genome phylogenies of bee gut microbiota phylotypes and SDP validation. **a–f** The upper part of each panel shows the maximum-likelihood core genome phylogeny inferred for each of the five core phylotypes of the honey bee gut microbiota and the non-core phylotype *B. apis*, based on the genomes in the reference database. Different shades of color represent candidate SDPs. The lower part of each panel shows the validation of the existence of SDPs using the metagenomic data. Orthologous sequences of core genes were extracted from de novo metagenome assemblies and aligned to the core genes of the reference genomes. The two density distributions correspond to the maximum alignment percentage identity of all extracted orthologs to their highest and second-highest (grey color) scoring candidate SDP. Note that only one distribution is shown for *S. alvi*, since no candidate SDPs were found in the database. The total number of non-redundant orthologous sequences extracted per phylotype is indicated in the legend of the density distribution plots. Phylogenetic analyses were based on nucleotide alignments, except in the case of Firm4. Bars correspond to 0.05 substitutions per site. Published species names are placed at the leaves of the trees (for full names, see Supplementary Data 1). Names of confirmed SDPs are shown next to the internal node corresponding to their most recent common ancestor. Results for the two non-core members *Commensalibacter* sp. and *F. perrara* are shown in Supplementary Fig. 6a. Larger trees with annotations can be found in Supplementary Fig. 5

hits to phylotypes of the honey bee gut microbiota in the non-redundant database of NCBI, most frequently to *Snodgrassella* and *Bartonella*, but some hits to *Gilliamella* were also observed (Supplementary Data 2).

In summary, a total of 11 SDPs were identified and validated within four of the five core phylotypes of the honey bee gut microbiota. This means that most core phylotypes of the bee gut microbiota have further diversified into genetically congruent units that can be detected with shotgun metagenomics. Compared to previously published species names, eight of the SDPs contain a single named species, two SDPs contain two species names, and one SDP has no associated species name yet (Fig. 2, Supplementary Data 1). While additional SDPs may be present at low abundance, our analysis shows that we have identified the most dominant SDPs, and, that these are present in bees in both Europe and the United States.

**Replication and stability of the microbiota changes with age.** Having confirmed the existence of SDPs within phylotypes, we next investigated whether there are age- or colony-specific signatures in the distribution of phylotypes and SDPs among individual honey bees. Both phylotype and SDP abundances were quantified based on the summed gene coverage of core gene family orthologs from the database (867–1,737 families per phylotype, see methods and Supplementary Fig. 7). For the seven phylotypes represented by complete genomes or genome scaffolds, a pattern consistent with replicating bacteria[29] was observed in the vast majority of samples (Supplementary Fig. 8), independently of the intra-sample phylotype abundance (Supplementary Fig. 9). Therefore, quantifications were based on the coverage at the estimated terminus in samples with detectable replication.

First of all, we found that the bacterial community had a highly stable phylotype composition, also at the level of individual bees,

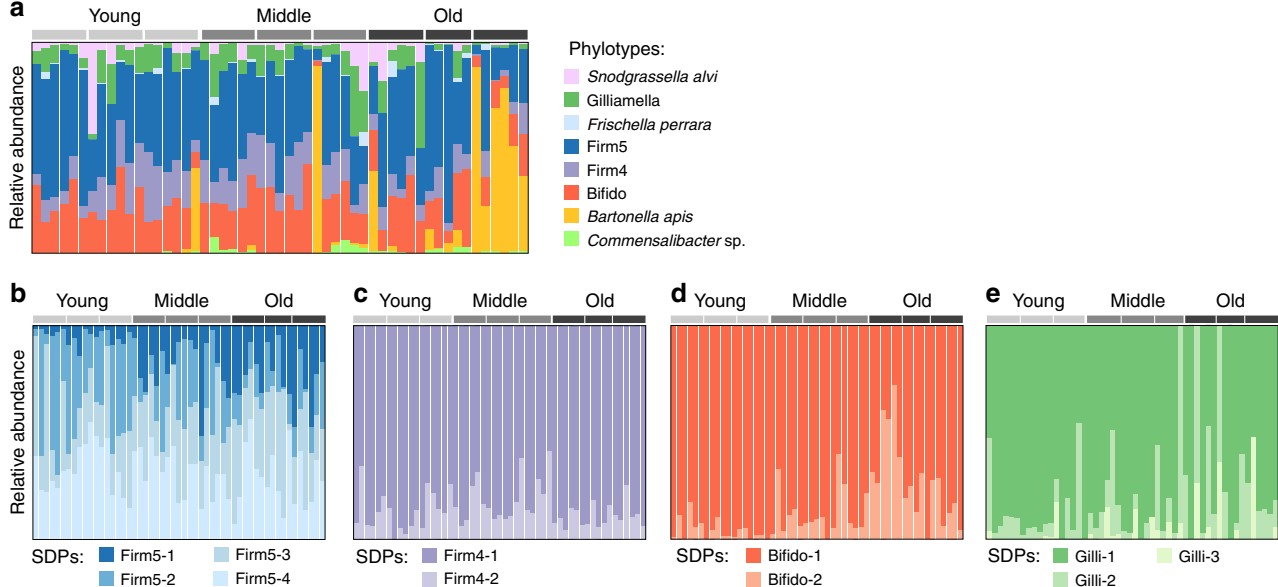

**Fig. 3** Community composition across samples at the phylotype- and SDP-level. **a** Composition at the phylotype-level. **b–e** Composition at the SDP-level for the four core members with validated SDPs. Abundance of a phylotype/SDP was set to zero if >20% of the core gene families used for quantification had a mean read coverage <1. Bee age group and colony origin are indicated on top of the graphs by grey bars. For each age group, the first and second bar indicate bees sampled from colony 1 in year 1 and 2, respectively, while the third bar indicates bees from colony 2

consistent with previous 16S rRNA-based analyses[13,15]. The five core phylotypes were detected in more than 94% of the samples, with Firm5 being the most abundant member (Fig. 3a, Supplementary Fig. 10). In contrast to most previous 16S rRNA-based studies, the Bifido phylotype was found to be the second most abundant. This difference is likely explained by the known 16S rRNA primer biases[30], indicating that the abundance of this member has so far been underestimated. The non-core species *Frischella perrara*, *B. apis* and *Commensalibacter* sp. were more variably present, with detected frequencies ranging from 0.37 to 0.68 across samples, also largely consistent with previous studies[13,31]. All other species included in our reference database had <1x genome coverage in any of the samples, and were therefore not analyzed further.

Overall, there was no strong association between honey bee age or colony and the relative abundance of core phylotypes in the current study (Supplementary Table 1). Most of the parameter deviance in phylotype abundance was contributed by the non-core members *Commensalibacter* sp. (ManyGLM, $P = 0.007$, 11.9%) and *B. apis* (ManyGLM, $P = 0.002$, 12.8%) (Supplementary Table 1), with *Commensalibacter* sp. being more frequently encountered in old bees and *B. apis* being highly dominant in one set of old bees (Supplementary Fig. 10), suggesting that old bees may be more prone to colonization by non-core members. Indeed, we observed an age-dependent increase in the fraction of unmapped non-host reads (Fig. 4a), and universal core ortholog sequences without a close hit to the current genomic database were also more commonly found in old bees (2/18 young bees, 8/18 middle-aged bees, and 15/18 old bees) (Supplementary Data 2).

These results not only show that old bees are colonized more frequently by non-core members, but also suggest that they exhibit higher variability in microbiota composition. To formally test this, we clustered the metagenomic orthologs of the 33 universal gene families at 95% nucleotide identity (analogous to 16S rRNA clustering), and recorded their occurrence across samples (see "Methods" section, Supplementary Fig. 11). Notably, the dispersion of beta-diversity was found to differ significantly between age groups, with the highest values found in old bees

(Fig. 4b). Since beta-diversity was calculated on presence-absence data, and the core members were present in essentially all bees, the increased beta-diversity must be due to the variable presence of non-core members (rather than a specific community not represented in the database). Intriguingly, the average population replication (PTR) for all core phylotypes also decreased with age (Fig. 4c, d), possibly rendering the community more susceptible to invasion by non-core members. Taken together, these results point in the direction of a decrease in gut microbiota stability with age.

At the level of SDPs, a high degree of co-occurrence was found within individual bees. For example, the four SDPs of the Firm5 phylotype were found to co-occur in all bees (Fig. 3b). Likewise, the two SDPs of the Firm4 and the Bifido phylotype co-occurred in the majority of samples, although in both cases one of the SDP was dominant across samples (Fig. 3c, d). In contrast, the distribution of SDPs of Gilliamella was more variable, with Gilli-2 and Gilli-3 only being present in a subset of samples (Fig. 3e). Interestingly, although Gilli-1 was generally dominant, it was completely replaced by the other SDPs in 3/53 samples. Thus, while most bees harbored multiple SDPs, the pattern of co-occurrence differed among phylotypes, suggesting that there are differences in the underlying mechanisms of co-existence. Moreover, despite an overall high degree of co-occurrence between SDPs within bees, subtle, yet significant, changes in relative abundances of SDPs with honey bee age were found for Firm4–2, Firm5-1 and Bifido-2 (ManyGLM, $p < 0.01$, Supplementary Table 2).

In summary, we find that SDPs of most phylotypes co-occur in individual bees, suggesting that they occupy distinct ecological niches. However, some of these SDPs display shifts in relative abundance according to age group and several lines of evidence suggest that the overall stability of the community decreases with age.

**Strains within SDPs segregate into individual bees.** As indicated by many long branches at the tips of the core phylogenies (Fig. 2)

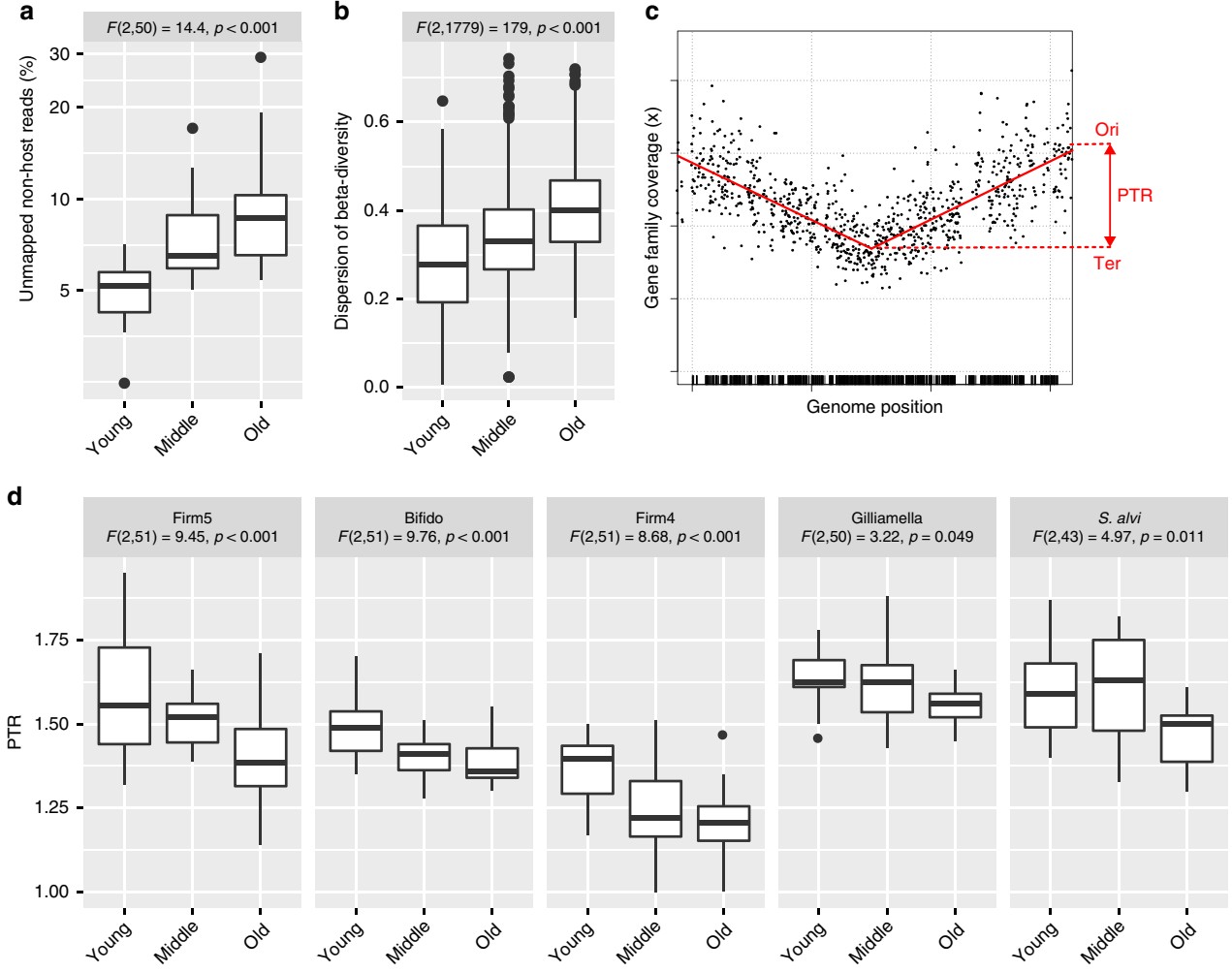

**Fig. 4** Decreased stability and average population replication in old bees. **a** Fraction of unmapped non-host reads (out of total reads not mapped to the honey bee genome). **b** Distance to the estimated centroid per age-group (using pairwise Jaccard distances between samples, calculated on data from 33 universal core gene families extracted from all metagenomic samples). **c** A representative plot showing the mapped coverage of core gene families relative to their location in a reference genome (all plots are provided in Supplementary Fig. 8). These plots were used to quantify the abundance of phylotypes/ SDPs in the metagenomic samples, but also to estimate the average population replication based on the ratio of the coverage at the origin and the terminus (peak-to-trough ratio, PTR). The red line corresponds to the segmented linear regression line that was fitted to the data to determine the mean coverage at origin and terminus. **d** Distribution of PTR values across the sampled bees for the five core phylotypes (based on samples with at least 10x terminus coverage). For each boxplot, the center line displays the median, the boxes correspond to the 25th and 75th percentiles, and whiskers extend to the most extreme data points that are within the 1.5 interquartile range of the box. For **a**, **b**, **d**, the results of the one-way analysis of variance (ANOVA) in response to age are shown above each plot. For panel **a**, the data was log10-transformed prior to ANOVA, and the plot is shown on a logarithmic scale

and the pairwise ANI values (Supplementary Fig. 4), strains of the same SDP can be highly divergent. Consequently, bees colonized by the same SDP could in theory harbor very different strains of that SDP. However, the genomes in the database were obtained from different geographic locations across Europe and the US, which may explain some of the observed intra-SDP divergence in the reference database. Given that all the bees sampled in the current study originated from two adjacent colonies, we therefore asked: do bees from the same colony harbor similar strains?

Firstly, the fraction of recruited metagenomic reads per genome varied substantially across samples (Fig. 5a, Supplementary Fig. 12), suggesting that individual bees do carry different strains. For example, three profiles were visible for Gilli-1 (Fig. 5a), which correlated with the core phylogeny, consistent with competitive exclusion among divergent strains within this SDP. In contrast, the profiles did not correlate with either sampling time or colony, nor did the geographic origin of the sequenced genomes appear to matter. In fact, for Bifido-1,

the four genomes that were isolated from the same location as the metagenomic samples recruited fewer reads than the genomes isolated elsewhere (Supplementary Fig. 12b).

To obtain quantitative measures of intra-SDP diversity, we inferred SNVs (single-nucleotide variants) based on a reduced database containing a single reference genome per SDP (Supplementary Fig. 13, Supplementary Data 1). The total fraction of polymorphic sites in the core gene sequences ranged from 4 to 33%, demonstrating a remarkably high level of strain diversity overall, but also large differences between SDPs (Fig. 5b). The most diverse SDPs were Bifido-1, Firm5-4 and *B. apis*, which were also highly diverse in the genomic database (Fig. 2, Supplementary Fig. 4). Within bees, the fraction of polymorphic sites was considerably lower, ranging from 0–11%, suggesting that strains compete for colonization or exclude each other after establishment within hosts (Fig. 5b). Indeed, most of the SNVs occurred at both high (>0.9) and low (<0.1) intra-sample relative abundances, indicating that strains can be both dominant and

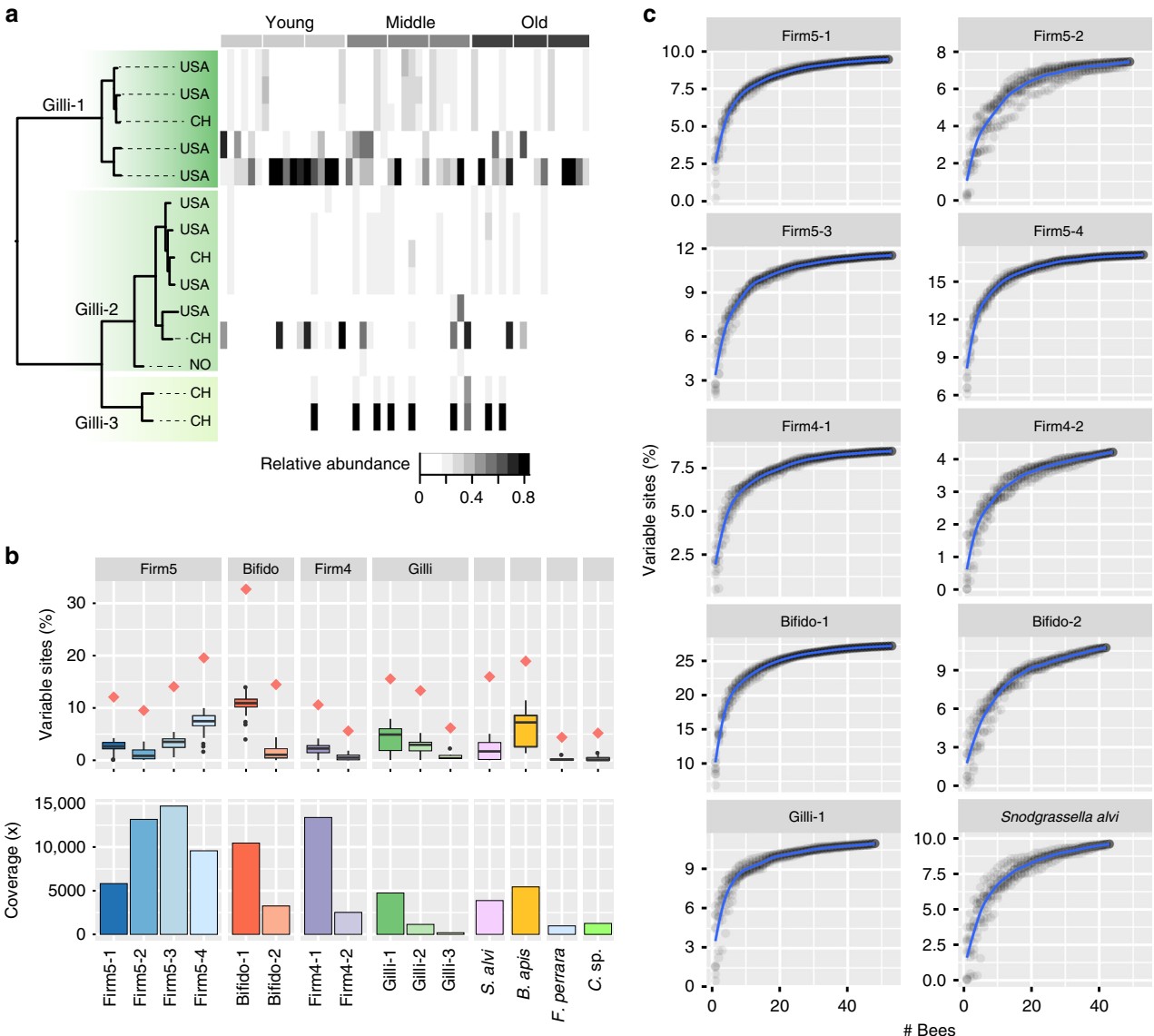

**Fig. 5** Strain-level diversity differs between SDPs, and strains segregate into different bees. **a** Proportion of metagenomic reads mapping to core genes in reference database genomes within each SDP of the phylotype Gilliamella. Each vertical bar in the heatmap corresponds to a single sample. The phylogenetic tree is the same as in Fig. 2. Node labels indicate geographic origin of genomes (CH: Switzerland, USA: United States of America, NO: Norway). **b** The upper part of the panel shows the percentage of polymorphic sites per SDP/phylotype within samples (the center line displays the median, the boxes correspond to the 25th and 75th percentiles, and whiskers extend to the most extreme data-points that are within the 1.5 interquartile range of the box), and the total fraction across the study is indicated by red diamonds. The lower part of the panel shows the total mapped coverage for each SDP/phylotype on the reduced database, across the study. **c** For the 10 SDPs/phylotypes with the highest representation across samples, the cumulative fraction of variable sites relative to the number of sampled bees is shown. Ten random sampling orders were done per SDP/phylotype, where the blue curve represents the smoothed conditional mean (loess, span = 0.2), and data for individual sampling orders are shown as grey circles

rare/absent across bees sampled from the same apiary (Supplementary Fig. 14).

The large fraction of variable sites observed, and the comparatively lower levels found within individual honey bees, prompted us to investigate how many bees would be needed to reach the diversity levels found across the study. By calculating cumulative curves of diversity, we found that approximately 20 bees were needed before the curves started to level off, regardless of the total diversity associated with a given SDP (Fig. 5c). Moreover, for each SDP, different sampling orders generated highly similar curves, suggesting that individual bees contain similar levels of strain diversity.

Bees in the current study were sampled to vary in collection time, colony and age (Fig. 1b), and these factors could potentially contribute to the total diversity observed. We therefore calculated the fraction of polymorphic sites corresponding to each of the nine samplings (Fig. 1b), and compared these values to the fractions obtained from random subsets of bees (Supplementary Fig. 15). While there was some variation in the fraction of variable sites for both the real and random samplings, there were no significant differences between them for any of the SDPs with sufficient representation for the analysis. This result therefore suggests that the high strain-level diversity found across the current study cannot be attributed to variation in age, sampling time, or colony affiliation.

To obtain a visual representation of the distribution of SNVs across individuals, we calculated the Jaccard distances between all

pairs of samples based on shared SNVs (see methods). Jaccard distances generally varied from 0.2–0.8, indicating that some bees do carry more similar strains than others. Principal coordinates analysis revealed well-separated clusters for some SDPs (e.g. Supplementary Fig. 16i, l). However, these clusters did not correlate with either colony origin or sampling time, consistent with the read recruitment patterns (Fig. 5a, Supplementary Fig. 12).

Taken together, these results demonstrate that the SDPs of the honey bee gut microbiota harbor very high levels of strain diversity, with competitive exclusion occurring at the level of individual bees, but not within colonies, resulting in individualized gut microbiota profiles. Intriguingly, individual bees seem to carry a similar fraction of the total strain-level diversity, regardless of the total diversity associated with the SDP.

**Functional variation of the microbiota across individual bees**. Having found that divergent strains coexist within colonies, but tend to segregate into individual bees, we hypothesized that individual bees would differ in the functional gene content of the microbiota, despite a stable composition at the phylotype level. Indeed, previous studies have uncovered a remarkably high level of gene content variation between strains belonging to the same phylotype[19–21]. However, the distribution of this diversity among individual honey bees has never been addressed.

To this end, we analyzed the distribution of 16,049 orthologous gene families predicted from our genomic database across the 53 bees for which we had sufficient shotgun metagenomic data. Since the detection of gene families is likely to be influenced by sequencing depth, we first normalized the abundance of all gene families relative to the abundance of the phylotype they were derived from. Following coverage normalization, gene families occurring with a relative abundance of <10% compared to the phylotype abundance in at least one sample were categorized as 'variably associated with the phylotype' (see methods and Supplementary Fig. 17). On the basis of this conservative threshold, a third of the gene families in the current database (5,059; 32%) displayed variable phylotype association (Fig. 6a, Supplementary Data 3). In the following, we will refer to these gene families as the 'variome'.

Given the existence of SDPs within phylotypes, gene families assigned to the variome might be expected to correspond largely to SDP-specific core gene content. However, many of the variome gene families contain members from multiple SDPs within the database, and only a small fraction of the gene families were found to correlate in abundance with a single SDP across the metagenomic samples (Fig. 6a, Supplementary 3), indicating that most of the variome does not represent SDP-specific core gene content. Besides many hypothetical or poorly characterized gene families, functions related to carbohydrate and amino acid metabolism and transfer (COGs 'G' and 'E') were prevalent in the variome (Fig. 6b). In particular, we found transporters and hydrolases for the utilization of dietary glycans to dominate a large fraction of the variome, especially in the core phylotypes Bifido, Firm4, Firm5, and Gilliamella. However, genes coding for cell surface structures or Type VI secretion systems were also abundantly present (Supplementary Data 3).

To gain further insights into the putative origin of gene families assigned to the variome, we clustered the genes of all variome gene families within their genomes of origin (see methods and Supplementary Fig. 17). On the basis of this analysis, between 49 and 73% of the variome gene families were found to occur in genomic islands (>10 kb and >5 genes) within the database (Supplementary Data 4). While a large fraction of these islands appeared to be of phage origin, many others coded for metabolic functions and cell surface structures (Supplementary Data 4). Islands

occurred at different frequencies across bees, no matter which broader functional category they belonged to (Fig. 6c). Interestingly, many of the intermediate or low frequency islands seem to be associated with specific strains rather than SDPs (Supplementary Data 3). For example, the island 'JG29_28' (phylotype Firm4), which encodes genes for ethanolamine catabolism (Fig. 6d)[32], occurred with a frequency of only 63% across samples, despite being associated with Firm4-1 (in the database), an SDP that is abundantly present in every bee (Supplementary Data 3–4, Fig. 3c). Ethanolamine is a membrane-derived compound that is prevalent in the gastrointestinal tract due to the high turnover of bacteria and host cells[32]. In the mammalian gut, pathogenic bacteria have been shown to utilize ethanolamine[33]. While its role in insects is currently unknown, ethanolamine utilization may provide a growth advantage in the bee gut depending on dietary conditions, state of the microbiota and fitness of the host. Many other islands with metabolic functions were linked to carbohydrate utilization. For example, a large island identified in the Bifido-1 SDP encoded key functions for the degradation and uptake of cellulose and/or hemicellulose (Fig. 5e), and was found to be present in only 90% of the sampled bees.

Metabolic flexibility encoded via the variome could potentially allow bees to adapt to changing environmental conditions or age-specific dietary requirements, while maintaining a stable composition at the phylotype level. Although we found no evidence for higher similarity in strain diversity for bees sampled together (Supplementary Fig. 15), functional redundancy among strains could potentially generate similar functional profiles from different strain compositions. To address this question, we quantified the fraction of the variome shared between all pairs of bees, with a gene family considered to be present in a sample when having a mean coverage of at least 10% of the phylotype coverage (see methods and Supplementary Fig. 17). The corresponding Jaccard distances mostly varied between 0.25 and 0.75 (Fig. 6f), indicating that while no pair of bees shared the same subset of the variome, some bees did share a larger fraction than others. Based on the Jaccard distances, significant changes related to age were found for the phylotypes Firm5 ($F(2,46) = 2,21$, $p < 0.001$, $R2 = 0.083$), Bifido ($F(2,46) = 2.27$, $p = 0.002$) and Firm4 ($F(2,46) = 3.23$, $p = 0.008$, $R2 = 0.117$), but not for Gilliamella ($F(2,45) = 1.66$, $p = 0.076$) and S. alvi ($F(2,38) = 1.17$, $p = 0.25$), which was also consistent with the principal coordinates analysis (Fig. 6g, h, Supplementary Fig. 18). However, as for the age-related changes in SDP abundance, the amount of the variability explained was rather low, indicating that additional factors are likely involved in shaping the functional profile of the gut microbiota.

In summary, we found age-related changes in the distribution of the variome for three out of five core phylotypes, despite the variability in strain composition among bees of the same age, indicative of functional redundancy across strains. However, pairs of bees rarely shared >70% of the variome, suggesting that the segregation of strains across bees results in functionally distinct bacterial communities at the level of individual bees.

## Discussion

Analyzing the distribution of genomic diversity in microbial communities is challenging due to the high complexity of most communities and the difficulty of sampling these communities in a consistent and controlled manner[34]. Taking advantage of the low phylotype-level diversity of the honey bee gut microbiota, we employed shotgun metagenomics to analyze the genomic diversity of this community at great depth. We systematically categorized the genomic diversity at three different levels (phylotype, SDP, and strain), profiting from a highly representative reference database, and investigated its distribution across genetically

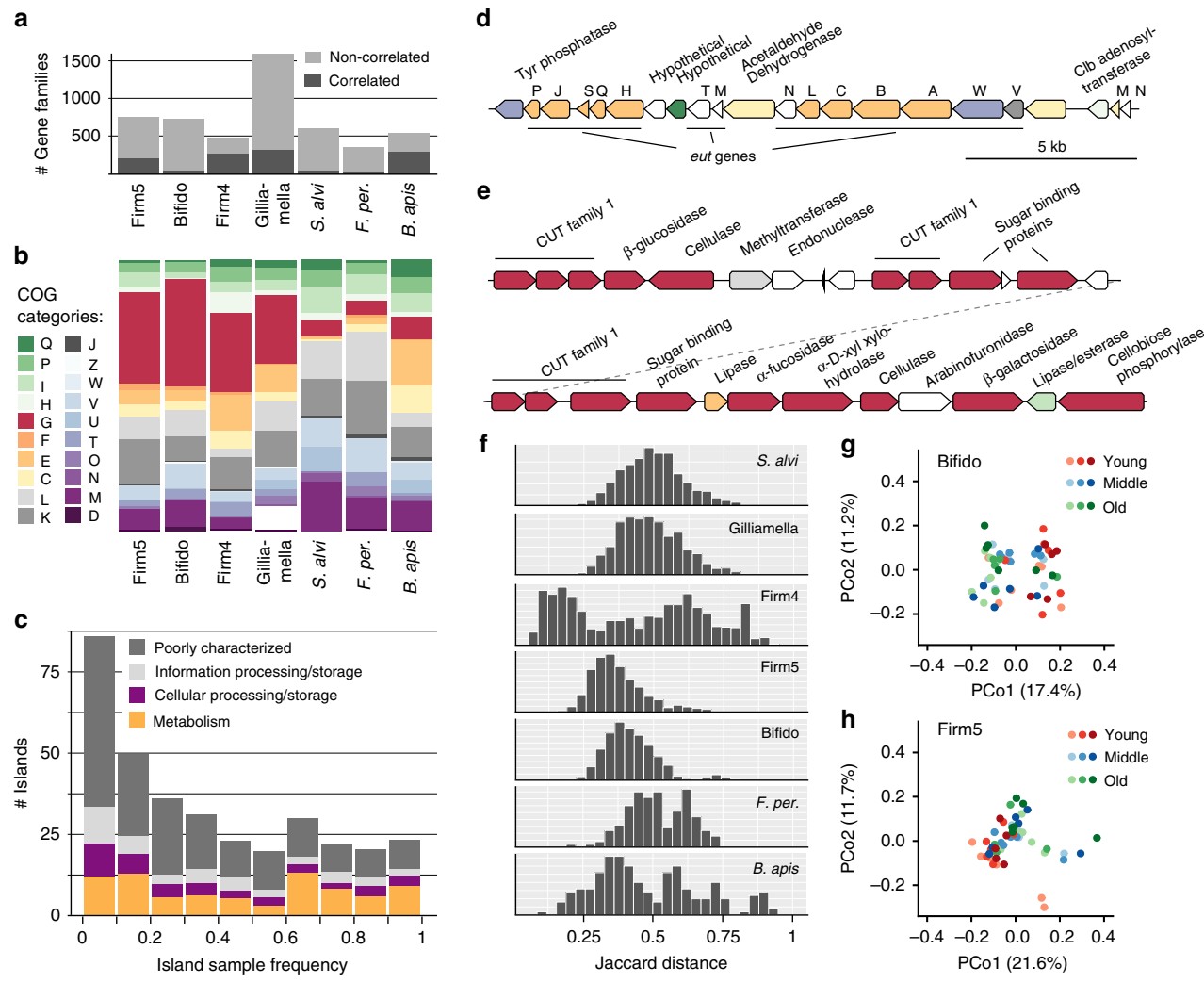

**Fig. 6** Functional gene content of the gut microbiota varies across bees. **a** Number of gene families per phylotype assigned to the variome (gene families occurring with a relative abundance of less than 10% compared to the phylotype abundance in at least one sample). The fraction of the variome correlating in abundance with a single SDP (or phylotype, in the absence of SDPs) is indicated in dark grey shade. **b** COG category distribution of variome gene families. Gene families having no COG category or belonging to category S (57% of all variome gene families) were excluded from the stacked bar plots. **c** Frequency of occurrence of genomic islands across the sampled bees. Within each island occurrence bin, the relative fraction of gene families corresponding to major COG classes is indicated. **d** Genetic organization of the part of island 'JG29_28' (phylotype Firm4) that encodes genes for ethanolamine catabolism. **e** Genetic organization of island 'Ga0133551_9 ' (phylotype Bifido), which encodes genes for cellulose and hemicellulose utilization. Gene colors correspond to COG categories as shown in (**b**). **f** Similarity between pairs of bees based on the shared variome, expressed as Jaccard distances with lower values indicating higher similarity. **g**, **h** Principal coordinates analysis of jaccard distances calculated from the variome, for the phylotypes Bifido (**g**) and Firm5 (**h**). Different shades of each color indicate colony origin and sampling year combination. PCoA plots for other phylotypes are shown in Supplementary Fig. 18

related individuals within and across colonies and different age groups.

First, we found that most phylotypes of the bee gut microbiota have split into SDPs, i.e. discrete evolutionary lineages that in most cases coexist in individual bees. Second, each SDP harbors further genomic diversity at the level of strains, segregating into individual bees. Third, while some age-related compositional characteristics were identified at the phylotype- and SDP-level, no correlations with age or colony appear to exist at the strain-level. And fourth, consistent with the segregation of strains into individual bees, we also found that bees differ considerably in the functional composition of their microbiota.

The presence of SDPs within core members means that the number of genetically coherent units in the bee gut microbiota is larger than what has been estimated on the basis of 16S rRNA amplicon sequencing studies. Moreover, several lines of evidence

suggest that the identified SDPs do not represent a local population structure, but are widely distributed. Notably, we re-analyzed the metagenomic data of a previous study from the U.S. and found that all SDPs were present and discrete. Moreover, in the reference database, most SDPs are represented by multiple genomes that originated from different geographic locations, and we found no evidence for increased read recruitment to genomes from locally isolated strains.

Several previous metagenomic studies have provided evidence for the existence of SDPs in natural bacterial communities, and a more recent study suggested that the divergence within and between such populations may conform to universal thresholds[11]. Specifically, Jain et al.[11] identified a 'discontinuity zone' of pairwise average nucleotide identity (ANI) in the 83–95% range between most publicly available genomes. Interestingly, the divergence within SDPs of the bee gut microbiota was in several

cases larger than the expected 95% ANI threshold within the genomic database (Gilli-1, Gilli-2, Firm5-4, Bifido-1). In fact, in the most extreme case (Bifido-1), nearly all of the pairwise ANI values were within the expected discontinuity zone. Moreover, when examining the density distribution of similarity between the core gene sequences extracted from the de novo metagenome assemblies and those in the current database (Fig. 2c), a considerable fraction fell within the 90–95% zone, indicating that additional equally divergent strains of Bifido-1 are present in the community.

High levels of diversity within SDPs were confirmed from the metagenomic data. Based on SNV profiling, the median fraction of polymorphic sites in the metagenomic data was 13% per SDP, with Bifido-1 being an extreme outlier (33% polymorphic sites). For comparison, a similar analysis on metagenomic samples from the human gut reported only 3.1% polymorphic sites in the 101 most prevalent species, across the entire dataset (207 individuals)[35]. However, while the read coverage was normalized to 10× (as in the current study), the cumulative coverage per SDP in the current study was orders of magnitude higher than in[35] (Fig. 5b), resulting in an overall deeper sampling of each SDP. Indeed, ref. [35] found that SNV discovery increased steeply until around 1000× (depending on the species), a coverage which is rarely obtained in metagenomic studies. Therefore, in the absence of other studies with a comparable sequencing depth per SDP, it is currently not possible to say whether the high strain-level diversity observed might also be common in other natural communities such as the human gut microbiota. For now, given the high levels of polymorphism, we can conclude that none of the sampled SDPs have undergone a recent selective sweep[10]. On the contrary, it appears that divergent strains of the same SDP can co-exist in the same geographic location, and even within the same colony.

These results raise several important questions: How does such high levels of diversity evolve, and how can it be maintained within the same host population? Based on how the diversity distributes among colony members, it appears that the answer depends on the taxonomic level of diversity considered (Fig. 1a).

At the SDP-level, niche differentiation seems plausible, considering their widespread co-occurrence within bees. Some significant changes in response to age were observed, both in terms of relative abundance of SDPs and functional profiles, suggesting that niche differentiation may be partly age-dependent. However, the amount of variation explained was modest, and a broader sampling (including more colonies from different geographic origins) is needed to confirm these patterns. In fact, the most conspicuous age-related change in the current study was a decrease in the average population replication (PTR) and the relative abundance of core members compared to non-core members, suggesting that the community may become unstable in old bees.

At the strain-level, both the SNV profiling and variome analysis indicated that strains segregate among individual bees. This is remarkable when considering that honey bees live in large eusocial colonies with frequent social interactions that in theory should facilitate microbiota homogenization across individuals. Based on cumulative fractions of SNVs across bees (Fig. 6c), at least 20–30 individuals are needed before the total diversity is reached, confirming that the diversity within bees is substantially lower compared to the diversity across the colony. Notably, this segregation appeared to be independent of both age and colony origin. In fact, the strain profiles of bees of the same age and colony, sampled at the same time, were as different from each other as those of other bees (Supplementary Figs. 15, 16). These results therefore point in the direction of competitive exclusion among strains, rather than niche differentiation.

Previous studies have shown that newly emerged honey bees acquire their gut microbiota within hours, and that the gut is fully colonized approximately by day 6 post emergence[31,36]. However, as of yet, no longitudinal studies on the gut microbiota have been conducted, and it is therefore not known whether the strain-level composition changes over time within individual bees. If the community composition is determined by the timing and order of the host-bacterial encounters in newly emerged bees, the individualized profiles could represent random early-life events ('priority effects')[37]. Moreover, if new strains are not readily incorporated in the community of fully colonized bees, initial differences in colonization could play an important role in facilitating the maintenance of strain-level diversity, and could hinder the occurrence of selective sweeps (which would otherwise reduce strain-level diversity). These observations are consistent with neutral theories of biodiversity[38], which predict that communities of islands or other patchy ecosystems (here individual bees) can vary, if they are inoculated from different starting communities and if dispersal is small enough that initial differences cannot be compensated.

Currently, little is known about the distribution of strains in natural populations of other environments. Host-specific strain-level composition in gut bacterial communities have also been observed in the human gut microbiota[35], but the reason for the host-specificity remains unclear. Considering the current results, it appears that a long lifespan, or large variation in diet or host genotypes is not needed for individualized gut communities. In fact, these patterns need not even be host-related. A recent study investigating the distribution of strain-level diversity in an enhanced phosphorus removal reactor also found that bacterial populations between granules are more different than expected from the overall reactor diversity[34].

The origin and functional relevance of individualized profiles in honey bees represents an exciting future avenue of research. Notably, we found that a large fraction of the variome gene families occurred in genomic islands, suggesting that diversification by horizontal gene transfer among community members might be prevalent. From a functional perspective, given that a large fraction of the variome was found to encode metabolic functions, most of which did not correlate in abundance with specific SDPs, it seems likely that strains of the same SDP carry distinct metabolic functions. If so, the communities found within individual honey bees might also, by extension, encode distinct metabolic profiles.

In conclusion, aside from providing fundamental insights into the genomic structure of a host-associated microbial community, our study provides a framework for future studies to test the link between bee health, ecology and microbiota composition at unprecedented resolution.

## Methods

**Sample collection**. Age-controlled honey bees (*Apis mellifera carnica*) were sampled from the apiary of the Engel laboratory at the University of Lausanne towards the end of the summer season (starting in September) in two consecutive years (2015 and 2016). To this end, a single frame was collected from each target colony, adult bees were brushed off the frame, and the frame was kept in the laboratory incubator at 34 °C over-night. The following morning, newly emerged bees were tagged with a marker pen on their thorax, and re-introduced to the colony together with the frame. At defined time points, a total of 54 tagged bees were collected from the two target colonies as shown in Fig. 1b. In year 1, six tagged bees were collected from the colony 'Les Droites' at day 10 (28/9-2015), day 24 (12/10-2015), and day 158 (22/2-2016). In year 2, six tagged bees were collected from the colony 'Les Droites' and the neighboring colony 'Le Grammont' at day 10 (25/9-2016), day 22 (7/10-2016), and day 47–48 (1-2/11-2016). Host genotype of the colony 'Les Droites' was the same in 2015 and 2016, as the queen was not replaced.

**DNA extraction and sequencing**. After anesthetization by $CO_2$, the guts were pulled out from each of the collected bees. Midgut and malpighian tubules were

removed, after which the hindgut (consisting of ileum and rectum) were collected in bead-beating tubes with 1 ml PBS (kept on ice for the duration of the dissection). Gut tissue was homogenized with a bead-beater using glass-beads (0.75–1 mm) for 30 s at speed 6.0. A series of centrifugation and filtration steps were then carried out to enrich for bacterial cells in the sample relative to host cells/tissue/pollen, all in PBS at room temperature. First, the homogenate was centrifuged at 2500 r.p.m., 5 min, to remove debris, and the supernatant was collected into new eppendorf tubes. The samples were then centrifuged at 9000 r.p.m., 15 min, to pellet bacterial cells. The supernatant was removed, and the bacterial pellets were re-suspended in 800 μl PBS. The suspension was again centrifuged at 2500 r.p.m., 5 min, to remove additional host-cells/debris. Finally, the sample was passed through a 10 μm filter, to remove large particles (pollen, remaining debris), and centrifuged at 10,000 r.p.m. for 15 min to pellet bacterial cells.

DNA was extracted from the enriched bacterial pellet using a CTAB-based DNA extraction protocol. For each sample, the bacterial pellet was re-suspended in 485 μl CTAB lysis buffer (100 mM Tris-HCl, pH 8, 1.4 M NaCl, 20 mM EDTA, 2% w/v CTAB) with 1.3 μl β-mercaptoethanol and 13.3 μl proteinase K (10 mg/ml). The samples were transferred to bead-beating tubes with zirconia/silica beads (0.1 mm), and homogenized on a bead-beater for two times 90 s, at speed 6.0. Samples were incubated at 56° C over-night. 5 μl RNase (10 mg/ml) was added to each sample, followed by incubation at 37 °C for 1 h. Finally, the DNA was extracted with PCI (phenol-chloroform-isoamyl, 25:24:1) and precipitated with 1/10 vol NaOAc (3 M, pH 5.2) and 2.5 vol 96% Ethanol, with 6 μl glycogen (20 mg/ml) added as a DNA carrier. DNA pellets were re-suspended in 20 μl $H_2O$.

All samples were sequenced at the Lausanne Genomic Technologies facility (GTF). Sequencing libraries were prepared with the Nextera XT library kit (Illumina), according to manufacturers instructions. A total of $2 \times 100$ nt paired-end sequencing was done on an Illumina HiSeq 2500 instrument, with 6 samples multiplexed per lane. For 6 samples of year 1, an additional multiplexed lane was done to ensure similar coverage across samples. The quality of the raw-data was evaluated with FastQC. This analysis revealed the presence of the Nextera adapters for some samples, which were trimmed off with Trimmomatic[39]. Only reads for which both members of the read-pair passed the trimming with a minimum length of 40 bp were kept for downstream analysis.

**Construction of a genomic database.** For each member of the honey bee core gut microbiota, all available published genomes (as of 2017) were downloaded from genbank. Moreover, genomes of *Lactobacillus kunkeei*, *Lactobacillus apinorum* and an unknown gammaproteobacterium, also isolated from the honey bee, but not considered to be members of the core gut microbiota, were added. In addition to previously published genomes, strains from the Engel lab stock collection were selected for sequencing, based on screening of the 16S rRNA gene. In total, five strains of *Gilliamella* sp., five strains of *Bifidobacterium* sp., one strain of *S. alvi*, one strain of *F. perrara* and one strain of *Commensalibacter* sp. (phylotype group 'alpha2.1') were sequenced[40]. On the basis of this collection of genomes, the database was streamlined to reduce redundancy due to highly related strains (maximum 98.5% pairwise ANI), prioritizing the most complete genome assemblies. Finally, for some draft genome assemblies, manual curation of the contigs was done, including removal of small contigs and re-ordering of contigs. To enable downstream functional analysis, all genes in the database were re-annotated with the eggnog-mapper[41] and pfam[42]. A list of all genomes in the final database and their accession numbers is provided in Supplementary Data 1.

**Metagenome mapping, gene coverage and de novo assembly.** Paired-end reads were mapped to the honey bee gut microbiota database using 'bwa mem' with default settings[43], after which various post-mapping filters were explored. Since *L. kunkeei* was found to be absent from all samples, mapping against the *L. kunkeei* genome served as a useful indication of unspecific mapping and filtering efficiency. Filtering based on read alignment length was found to be superior compared to edit distance (which is a measure of the similarity between the read and reference in the aligned part of the read). Therefore, all bam-files were filtered so that only reads with a minimum alignment length of 50 bp were considered mapped. On the basis of the filtered BAM files, gene coverage data was generated for all genes in the database.

First, BED files (indicating gene locations within genomes) were generated from the genome annotations. To reduce variation due to the use of different annotation pipelines, and noise related to mapping against short regions, genes shorter than 300 bp were removed. Second, the number of mapped reads per gene was counted with samtools[44], based on which the gene coverage was calculated (average read coverage per bp).

To infer de novo metagenome assemblies, paired-end reads were first mapped against the honey bee genome (using bwa mem with default settings), to filter off host-derived reads. Unmapped reads were extracted with samtools (flag -f 4) and assembled independently for each metagenome sample, using the SPAdes metagenome assembler (v.3.9.0) with default settings[45]. For each assembly, the resulting contig file was filtered to exclude contigs shorter than 500 bp, or with a k-mer coverage <10. Putative ORFs were predicted with Prodigal v.2.6.3[46] using the metagenome flag (-p meta). ORFs shorter than 300 bp were excluded from downstream analysis.

**Database coverage.** To investigate the extent of microbial diversity not represented in our current database, reads not mapping to the database (minimum alignment length 50 bp) were mapped against the honey bee genome, using bwa mem as previously, and each of the fractions were counted with samtools. Furthermore, protein sequences corresponding to 33 universal orthologs were extracted from the ORFs predicted on each of the metagenomic assemblies, using FetchMGs03.pl from the MOCAT package[28]. All extracted universal ortholog sequences were first blasted against the honey bee gut microbiota database, and the subset without a close hit to the genomic database was extracted (BLAST nucleotide alignment identity lower than 95%). For the extracted subset, the total number of sequences and the median number per universal gene family was calculated for each sample, and blasted against the non-redundant database of NCBI. The putative taxonomic affiliation of lineages not represented in the database was determined for each sample, based on the most frequent hits, using the BLAST taxonomy database.

**Discovery and validation SDPs.** Orthologous gene families were predicted separately for the genomes of each phylotype in the database using Orthofinder[47], and gene families corresponding to single-copy core genes were extracted. For *Commensalibacter* sp., for which only a single genome sequence was available, two draft genomes with 96% sequence identity in the 16S rRNA gene compared to the honey bee-derived isolate (ATSX01000000, AGFR01000000) were included for the ortholog prediction only, in order to generate a set of putative core genes. Core genome phylogenies were inferred individually for each phylotype represented by at least 3 strains. For each core gene family, the sequences were aligned at the protein level with mafft[48], back-translated to nucleotides and trimmed for columns represented by less than 50% of all sequences. Core genome phylogenies were inferred on the concatenated trimmed alignments, using RAxML[49] with the GTRCAT model and 100 bootstrap replicates. For the Firm4 phylotype, which is represented by two genomes in the current database, a phylogeny was inferred by including 4 unpublished genomes, without back-translation to nucleotides, using the PROTCATWAG model in RAxML. Pairwise ANI values were calculated all-against-all, using FastANI[11]. After inspection of the phylogenies and ANI tables, candidate SDPs were identified as forming discrete clades with 100% bootstrap support, and having a minimum pairwise ANI of 89% within clusters. To reduce noise in downstream analysis related to specific gene families with a high level of similarity across SDPs (due to i.e. conservation or recent homologous recombination), the core gene families of each phylotype harboring candidate SDPs were further filtered to exclude any families with more than 95% nucleotide alignment identity in any pairwise comparison across SDPs.

To validate the candidate SDPs, orthologs of all filtered core gene families were extracted from the de novo metagenome assemblies with blastn, using the core gene sequences in the database as queries. Specifically, a metagenomic ORF was considered an ortholog for a core gene family in the database if the blast hit had an $e$ value < 1e-5, a minimum percentage id of 80%, and a query coverage >0.5. The clustering of metagenomic ORFs around the candidate SDPs was quantified based on percentage identity between the metagenomic ORFs and their orthologous core gene family sequences. For each of the extracted ORFs, the sequence was added to the corresponding core gene alignment with mafft (method 'add fragment'), resulting in an alignment containing the core gene family sequences from the database with a single metagenomic ORF. The percentage identity was calculated between the ORF and each gene in the alignment using the Bioperl SimpleAlign module. Finally, the maximum percentage identity in the alignment to each of the two highest-scoring SDPs was recorded for each ORF. For phylotypes without candidate SDPs, the same procedure was applied using the full set of core gene families, and reporting the maximum percentage identity in the database (see also flowchart in Supplementary Fig. 3).

**Quantification of community members.** For phylotypes without validated SDPs, the metagenomic read coverage of all the genes in each core family was summed. For phylotypes containing validated SDPs, the same procedure was applied per SDP, using the filtered core gene families. Next, a single genome was chosen as reference per phylotype/SDP, and the summed core gene family coverages were plotted according to their position in the reference genome. Since a pattern consistent with replicating bacteria was observed in the majority of samples for all the phylotypes/SDPs harboring complete genomes or scaffolds (genes located close to the origin of replication had higher coverage than genes located close to the terminus), a segmented linear regression line was fitted to the data, and the raw abundance was taken as the coverage at the estimated terminus (the intersection between the regression lines). Additionally, the peak-to-trough ratio (PTR) was calculated from the fit (maximum coverage at *Ori*, divided by coverage at *Ter*)[29]. For samples where the segmented lines did not show a proper fit (terminus location inferred far from estimated breakpoint, or coverage at *Ori* lower than *Ter*), the median coverage of all core gene families was used instead, and the PTR was set to 1. Finally, the raw abundance of a phylotype/SDP was set to zero if less than 80% of the core gene families used for quantification had a mean read coverage >1 (see also flowchart in Supplementary Fig. 7).

To address the question of whether old bees are more prone to colonization by variable non-core members than younger bees, the sequences of each of the 33 universal ortholog gene families extracted from the de novo metagenome

assemblies were first clustered at 95% nucleotide identity with uclust[50]. Second, for each cluster, the presence across samples was inferred based on the cluster members, and 'OTU'-tables with presence-absence data were constructed. Finally, the jaccard-distances between all sample pairs were calculated, and the distance to the centroid for each age group was estimated (using 'vegdist' and 'betadisper' functions from the R package 'vegan').

To obtain an overall result for all of the 33 universal gene-families, the 'distance-to-centroid' data obtained for each gene-family was extracted and used together for statistical testing (see also flowchart in Supplementary Fig. 11).

**Strain-level analysis.** The recruitment of reads to each genome was analyzed based on the number of reads mapped to the core genes of each genome. Starting with the total raw number of recruited reads per sample and genome (normalized for the total core gene length), the relative fraction of reads recruited per genome within each SDP was calculated and visualized using the R package 'heatmap.2'.

To quantify diversity within SDPs, reads were mapped against a reduced version of the database, containing one genome representative per SDP or, in the absence of validated SDPs, per phylotype (genomes listed in bold letters, in Supplementary Data 1) using 'bwa mem'. To reduce errors in SNV calling caused by eventual indel alignment errors, an edit distance < 5 was required for a read to be considered mapped, in addition to the minimum alignment length of 50 bp previously employed. Candidate SNVs were first called on the filtered bam-files using the metaSNV pipeline with default settings[51]. To limit the influence of variably present genes, only SNVs located within genes corresponding to the core gene families were included in the analysis. Given the exceptionally high coverage obtained in the current study, a second filtering of SNVs was done with the following cutoffs: (i) SNVs were only quantified in samples where the corresponding reference genome was present with at least 10x terminus coverage; (ii) for samples with sufficient coverage, SNVs found in genes with less than 10x mean coverage were treated as missing data; (iii) intra-sample relative SNV abundances <10% were set to zero (corresponding to the detection limit with 10x coverage). Following these steps, only SNVs remaining polymorphic were included in the analysis (not fixed across samples, and with a with an intra-sample relative abundance of at least 10% in at least one sample) (see also flowchart in Supplementary Fig. 13).

After filtering, the fraction of variable sites within core genes was calculated for each SDP, both per sample and across the full dataset. Furthermore, the cumulative increase in the fraction of variable sites relative to the number of bees sampled was calculated, using 10 random sampling orders per SDP. To investigate whether differences in sampling might contribute to the observed levels of diversity, the fraction of variable sites was first calculated for each sampling (9×, see Fig. 1b) for all taxa with sufficient coverage representation (at least 6 samplings with at least 5 bees having a terminus coverage >10×). Next, 27 random subsets of 6 bees were generated for the full dataset, and the fraction of variable sites in these subsets was likewise calculated.

To visualize the distribution of SNVs across samples, and identify eventual patterns related to honey bee age or colony, a distance matrix was generated based on shared polymorphic sites. Specifically, a SNV was considered to be shared between two samples if it occurred with an intra-sample relative abundance of at least 10% in both samples. The jaccard distance was calculated as the fraction of non-shared polymorphic sites divided by the number of sites scored in both samples. The resulting matrix was visualized with a principal coordinates analysis.

**Functional variation across metagenomic samples.** To investigate functional variation in the gut microbiota of individual bees, we analyzed the distribution of all orthologous gene families predicted from the genomes in the database for each phylotype. To reduce noise related to low coverage, only samples with a phylotype terminus coverage of at least 10x were included in the analysis of gene content variability. For functional inferences, all EggNOG and Pfam annotations were transferred from the gene annotations to the gene families.

The raw gene family coverage was calculated as the summed gene coverage for all genes within the family. To normalize for variation in coverage between samples and phylotypes, the coverage of each gene family was linearly re-scaled according to a phylotype terminus coverage of 10× in each sample. For example, a gene family with a coverage of 25× in a sample with a phylotype terminus coverage of 50x would result in a normalized gene family coverage of 5×. Following coverage normalization, gene families with less than 1x coverage in at least one sample were considered to be variably associated with the phylotype (referred to as the 'variome') (see also flowchart in Supplementary Fig. 17).

Since the variome was inferred at the phylotype level, variome gene families could potentially encode SDP core gene content, in which case their abundance is expected to correlate with the SDP abundance[52]. Therefore, to estimate the subset of the variome belonging to the SDP core gene content, pearson correlation coefficients were calculated between the raw abundances of the variome gene families (log10-transformed) and all SDPs (SDP terminus coverage, log10-transformed). Variome gene families with pearson correlation coefficients larger than 0.8 for a single SDP were considered to be correlated with the SDP[52].

To address the question of whether bees sampled from the same colony and/or time-point share a larger fraction of the variome than other bees within the study, a jaccard distance matrix was generated, where a gene family was considered to be present in a sample when having a normalized coverage of at least 1×.

To gain further information on the distribution of variome gene families within genomes, the genes assigned to the variome were clustered within their genome of origin. Specifically, genes with an intergenic distance of less than 5 kb were joined in an 'island'. 'Big islands' were defined as having a minimum length of 10 kb, and containing at least 5 genes. To avoid excessive redundancy due to identical islands being present in more than one genome in the database, islands for which all their associated gene families were also present in another island (complete overlap) were removed. Finally, the occurrence of islands across samples was recorded, where an island was considered to be present when at least 80% of the gene families contained in the island were present in the sample.

**Statistics.** To analyze variation in community composition, the raw abundances (see 'Quantification of community members') for all SDPs and phylotypes were converted to relative within-sample abundances, and analyzed with a generalized linear model using the negative binomial distribution (visually confirmed to normalize for the quadratic mean-variance relationship) and Age/Colony as explanatory variables[53,54].

Changes in the fraction on unmapped non-host reads (log10-transformed), dispersion in beta-diversity and average population replication (PTR) in response to age were tested using one-way analysis of variance (ANOVA). Differences in the mean fraction of variable sites (SNVs) in subsets of bees corresponding to samplings (9×, see Fig. 1b) versus random subsets of 6 bees were tested with Welch's $t$ test, applied on each comparison.

Changes in the distribution of the variome relative to age and colony affiliation were tested for each phylotype using the jaccard distance matrices (see 'Functional variation across metagenomic samples'), with PERMANOVA (function 'adonis', R package 'vegan').

**Code availability.** Documentations of the workflow, including all scripts, databases and result files, are available on Zenodo [https://doi.org/10.5281/zenodo.1479668][55]. All scripts were written in perl, bash or R. Statistics and plots were likewise done in R (using packages: ggplot2, gridExtra, reshape, gplots, RColorBrewer).

**Reporting Summary.** Further information on experimental design is available in the Nature Research Reporting Summary linked to this article.

## Data availability

The raw data has been deposited in the sequence read archive (SRA) with the accession SRP150166. Source data underlying the results shown in the figures of this manuscript are available on Zenodo [https://doi.org/10.5281/zenodo.147966][55].

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

## Acknowledgements

We would like to thank Diego Gonzalez, Fabienne Wichmann, German Bonilla-Rosso, and Laurent Keller for critical reading of the manuscript and discussions. This work was funded by the HFSP Young Investigator grant RGY0077/2016, the ERC-StG 'MicroBe-eOme', the Swiss National Science Foundation grant 31003A_160345, the University of Lausanne, and the Fondation Herbette at the University of Lausanne.

## Author contributions

K.M.E.: conceptualization, methodology, software, validation, formal analysis, investigation, methodology, resources, visualization, data curation, writing—original draft, writing—review & editing, P.E.: conceptualization, methodology, resources, validation, visualization, writing—original draft, writing—review & editing, project administration, supervision, funding acquisition.

## Additional information

**Competing interests:** The Authors declare no Competing Interests.

