## [Peer Review File · Nature Communications]

Reviewers' Comments:

Reviewer #1:

Remarks to the Author:

In this study Ellegaard and Engel analyze the microbial diversity of the honey bee gut microbiome using shotgun metagenomic sequence. The results provide a very fine-scale analysis of these bacteria and interesting insights on the composition of these microbes in their natural environment. Overall, I think the results are interesting and the methods seem thorough. However, I am not sure I completely understand all the steps of their approach and I also had trouble following the results. I think that the authors should clarify some of their analyses. Below I have listed some more specific comments:

It was not easy to follow the different steps of the methods. This is because some results were obtained either from i) the complete genomes of genbank, ii) read-mapping against these genomes or iii) from the de novo assembly of the reads. For this last category, it is also not clear whether only the unmapped reads were de novo aligned or whether the mapped reads were also de novo aligned. By making these steps more explicit, it would make the manuscript easier to follow.

I think that the authors provide robust evidence for the existence of the SDPs based on their phylogenetic trees. However, the SDPs are then defined with 89% ANI. This hard threshold sounds a bit arbitrary and is clearly below the common ANI thresholds used to define species. It could be interesting to define species based on the ANI thresholds typically used to delimitate species (around 95% identity). I am not saying that the authors should re-do their analyses with these thresholds, but it could be interesting to discuss how the commonly used species thresholds relate to the natural composition that the authors observe (their SDPs) in these well-sampled communities. It should also be explained why the authors decided to use the threshold of 89%.

The SDPs were defined based on the reference genomes found in public databanks. For what I understood (but I might be wrong), all the SDPs were represented in genbank. The metagenomic data analyzed by the authors confirmed the existence of these SDPs but didn't identify any new ones. It would be useful to state explicitly in the Discussion that the genomic diversity of these species in genbank is representative of the diversity observed in natural populations. If I misunderstood these results it would be useful to state which species are not well represented in genbank. This would help guide future sequencing efforts.

I am a little confused about the definition and analysis of the variome. From what I understand the variome is composed of all the genes present at less than 10% frequency of the core genome of each phylotype. The authors then look at the association of these frequencies to the frequency of the different SDPs. Isn't there a methodological flaw here? If some SDPs are found at 30% frequency in a phylotype, it is then impossible for any genes of the variome to be defined as a "core gene" of that SDP, since the genes of the variome can only reach 10% frequency (by definition). An alternative approach could be to look for the genes that are present in the same frequency as the genes of the SDPs and compare that to the random expectation (e.g. relative to random frequencies). This might help the authors identify whether there are SDP-specific genes.

The term "variome" was new to me, although other terms have been used to define the same notion (e.g. "the accessory genome" or "the flexible genome"). I don't think we need to introduce a new word that would add more confusion. Also, when googling "variome" it appears that this term is predominantly used to refer to the variable SNPs within the human genome, which is quite different from what the authors are talking about.

The authors wrote they removed very similar genomes from their databank, but they don't explain what criterion has been used to define the very similar genomes.

Reviewer #2:

Remarks to the Author:

The study by Ellegaard and Engel investigates the intra-population diversity among common microbial residents of the bee gut microbiota. For this, they use a reference genome collection for major 'phylotypes' of the bee gut (which includes novel genomes they sequenced from their own lab stock collection) to recruit reads from honey bee gut shotgun metagenomes, which are generated from individual bees that cover multiple age groups and colonies. The study is overall well-written and clear, figures and tables are well-prepared.

While I commend the authors for their excellence in generating such a valuable dataset which brings insights to many fundamental questions of microbiology within reach, I am left somewhat disappointed with their analysis of the data, which essentially led to rather uninteresting findings that can be summarized to this: 'we have culture representatives for the vast majority of the most abundant phylotypes of the bee gut, yet microbes of some bees are not identical to the cultured isolates we have'. I don't have the impression from their text that their analysis of these data addressed any of the important questions they postulated in their introduction.

The number and average identity of sequence-discrete populations given the reference collections can indeed tell us about the coverage of our isolates, which in almost every single environment leads to the same conclusion: metagenomes contain more diversity compared to what we managed to cultivate, and 16S rRNA gene amplicons are not suitable targets to characterize it. But this strictly reference-based approach does not tell us anything about 'why' it is the case, which is the part that is not yet established. With its superb experimental design the study gets very close to taking a stab at fundamental questions that could be relevant to the broader community of microbiologists, yet it falls short. These may be interesting findings for the bee community, but in my opinion the scope of findings do not go beyond that in my opinion as I don't see how recruiting reads from pooled individuals helps us see anything besides SDP curves that have no biological meaning. I believe this dataset could have revealed much more regarding the gut microbial evolutionary and ecological dynamics of bees within and across colonies if the authors reconstructed individual genomes from individual bees and use established practices of comparative genomics, which could shed light into the accessory genomic regions and how they compare to isolated representatives of phylotypes and to each other across different individuals.

Another surprise for me was the statement in line 438: "The genomic database, bed-files, gene sequences/identifiers, ortholog predictions, annotations, and key scripts will be made available in a public repository upon publication". By not making available their data and ad hoc scripts to reviewers, perhaps authors are suggesting that they do not trust reviewers' ability to not take their stuff and run away. I am not sure how reviewers are expected to distinguish fiction from science when data and scripts are not accessible for inspection. I find this attitude not only insulting, but also against the nature of scientific review. There are many services that would allow anonymous access to data and scripts without making them public, and "we will make them available after publication" should not be acceptable for credible journals that strive for excellence during the review process.

Besides these major points regarding the focus and the depth of the study, I would like to thank the authors for their meticulous data analyses. While I believe their strategies led to rather uninteresting

findings, high level of meticulousness for every stage of the analysis is quite impressive and exemplary.

Some minor points:

182: Reminiscent of dysbiosis? How do authors define dysbiosis, and based on what they link increased abundance of microbes that do not match to known phlotypes and decreasing replication rates for the ones that are known? Aging could be correlated with changes in diet and/or immune system functioning. Why is it necessary to link it to a buzzword that is not even properly defined and understood where it is studied most extensively? Overall this section seems disconnected from the title. The finding here is simply that the culture collections are more representative of younger individuals. I think the authors should consider defining what they mean by this term clearly and discuss it more carefully.

Another such word that makes a surprising appearance is 'speciation'. Speciation appears three times in the manuscript, one in each of the following sections: the abstract, results, and discussion. The results section is where the justification is needed the most, yet lacking. The authors associate differences in intra-SDP diversity among core genes of representative genomes through read recruitment with stages of speciation. Most microbiologists could agree with that statement, and I would have let it go if I had read it in a Tweet. But here a stronger justification could be beneficial.

Showing more examples of the underlying coverage data from which PTRs were assessed as supplementary figures could help. Because the observation of decreased coverage may be a by-product of decreasing coverage of the populations authors have access to. As authors also mentioned, Figure 4 shows decrease in coverage for almost all phylotypes and increase in unmapped reads in elder bees. I think it is important to solidify the correlation between decreasing coverage and PTR to avoid any potentially misleading claims.

306: I think the argument that there is a universally conserved make up of the bee gut microbiome beyond geographical distances is conceivable. But isn't it a bit too much to say in this level of conviction when the only additional data are coming from a single sample that pools a bunch of bees in the US? What is the genetic makeup of those bees in the US sample? Maybe being more cautionary would better serve everyone.

312: I do not agree with the authors' suggestion here. Genome-resolved metagenomics, phylogenomic and comparative genomic analyses of resulting population genomes would be much more suitable for attempts to delineate 'species', whatever that means. Many groups are already doing exactly that, demonstrating the efficacy of having genomes rather than counting single-nucleotide variants using references. What is done in this study is simply trying to address a very complex problem through crude associations. The approach used seems suitable to show how irrelevant / relevant isolate genomes are for a given metagenome, and even then it is not very useful because it is unable to discuss the presence of accessory genes that may be in the metagenome but not in isolate genomes, making it impossible to discuss whether any of the isolate genomes in the collection truly represents the functioning of the population to which it matches with confidence.

431: It may be the case for the bee community, but other groups that study marine and terrestrial ecosystems may find these techniques rather coarse. Perhaps there is no need for such qualitative statements since our level of resolution seems to be increasing rapidly in all fronts.

Point by point rebuttal

Reviewer #1 (Remarks to the Author):

In this study Ellegaard and Engel analyze the microbial diversity of the honey bee gut microbiome using shotgun metagenomic sequence. The results provide a very fine-scale analysis of these bacteria and interesting insights on the composition of these microbes in their natural environment. Overall, I think the results are interesting and the methods seem thorough. However, I am not sure I completely understand all the steps of their approach and I also had trouble following the results. I think that the authors should clarify some of their analyses. Below I have listed some more specific comments:

It was not easy to follow the different steps of the methods. This is because some results were obtained either from i) the complete genomes of genbank, ii) read-mapping against these genomes or iii) from the de novo assembly of the reads. For this last category, it is also not clear whether only the unmapped reads were de novo aligned or whether the mapped reads were also de novo aligned. By making these steps more explicit, it would make the manuscript easier to follow.

Reply: We agree with the reviewer that it was challenging to follow the different steps of the methods in the previous version of the manuscript. In the revised manuscript, we have therefore included six different flowcharts as supplementary Figures; Supplementary Fig. 1 gives an overview of the entire analysis, providing a rough outline describing how the input data (metagenomes or reference genomes) was processed and combined to obtain the five major types of output data (SDP validation, SDP abundance/replication, beta-diversity analysis, SNV analysis and variome analysis). For each of these five analyses, we additionally provide a more detailed flowchart (Supplementary Figs. 3, 6, 9, 11, 15), which should facilitate the understanding of the details of each analysis and show how the information from the reference database was integrated with the metagenomic data.

We also carefully revised the manuscript text in terms of the described methods and approaches with the aim to make the explanations more explicit and consistent. We hope that these edits, together with the new supplementary figures, have made our analyses more transparent and understandable. We thank the reviewer for pointing this out to us!

I think that the authors provide robust evidence for the existence of the SDPs based on their phylogenetic trees. However, the SDPs are then defined with 89% ANI. This hard threshold sounds a bit arbitrary and is clearly below the common ANI thresholds used to define species. It could be interesting to define species based on the ANI thresholds typically used to delimitate species (around 95% identity). I am not saying that the authors should re-do their analyses with these thresholds, but it could be interesting to discuss how the commonly used species thresholds relate to the natural composition that the authors observe (their SDPs) in these well-sampled communities. It should also be explained why the authors decided to use the threshold of 89%.

Reply: The reviewer raises an important point. ANI values were actually not used as a hard cut-off; rather, we looked for discrete clusters, based on the phylogenies and the ANI matrix, without making assumptions of the diversity within clusters (i.e. ANI within clusters). Most of the SDPs do in fact conform to the 95% threshold proposed to delineate species, but the Bifido-1 SDP turned out to be a rather extreme outlier. This is an interesting point in itself, because our analysis indicates that the strains within this SDP form a continuous population, without any obvious clusters. It is because of the exceptionally high diversity within this SDP (which is also observed at the SNV level) that we speculate about an on-going speciation/diversification process. We have now made a much more detailed discussion about this point (line 406-433), which we hope will clarify both how we define the SDPs, but also the possible implications of having SDPs falling outside the norm (i.e. the 95% ANI thresholds typically used to delineate species).

The SDPs were defined based on the reference genomes found in public databanks. For what I understood (but I might be wrong), all the SDPs were represented in genbank. The metagenomic data analyzed by the authors confirmed the existence of these SDPs but didn't identify any new ones. It would be useful to state explicitly in the Discussion that the genomic diversity of these species in genbank is representative of the diversity observed in natural populations. If I misunderstood these results it would be useful to state which species are not well represented in genbank. This would help guide future sequencing efforts.

Reply: It is correct that genomes of most SDPs were publicly available in genbank. However, a systematic analysis of these genomes to identify candidate SDPs and their validation by metagenomics has not been carried out before. Hence, our analysis shows, for the first time, that most of the previously identified strains

(for some of which species names have been proposed) represent sub-lineages that form sequence-discrete
populations (SDPs) in nature. In addition to the genomes in genbank, we have sequenced genomes
specifically for the purpose of this project. This has bolstered the phylogenetic analyses and increased the
genomic diversity present in the database. Moreover, it has led to the identification of one new sublineage
that we confirmed to represent another SDP (Gilli-3). All genomes are now available on Genbank, and their
accession numbers are provided in Supplementary Data 1 (we have published a Microbiology Genome
announcement recently).

We have revised the analysis concerning the database representation of SDPs, and the eventual presence
of other SDPs as specified here below:

First, we have added a new plot panel to Supplementary Fig. 2, which shows the number of mapped reads
as a fraction of the reads not mapping to the host (i.e. corresponding to the bacterial community, plus
eventual other sources, such as viruses, pollen-derived DNA etc). This plot shows more clearly that about
90% of the reads that could correspond to bacteria map to our database, indicating that the most abundant
SDPs should be represented. We have added a corresponding sentence that states this explicitly on line
105-109:

*"The enrichment protocol efficiently reduced host-derived DNA in most samples, with an average of 13% of*
*the reads mapping to the honey bee genome (Supplementary Fig. 2a). Moreover, approximately 90% of the*
*remaining reads mapped to the reference genome database in the majority of samples, regardless of honey*
*bee age, indicating that the current database is representative of most of the community (Supplementary*
*Fig. 2b)"*

Second, We have now extracted universal ortholog sequences from the full metagenomic assemblies (rather
than separate assemblies of unmapped reads), and analyzed the subset of the sequences without a close
hit to the database (95% nucleotide identity). This analysis does indicate that some phylotypes may harbor
additional SDPs, or at least divergent strains not present in the current database, e.g. for *Snodgrassella*
(Supplementary Data 2). Therefore, we have also revised the results (line 114-125), focusing more on the
SDPs that we have confirmed to be present, rather than how many there may be in total, or whether we
have found them all. However, as stated above, we are confident that the most abundant SDPs and most of
the genomic diversity is represented in our database, and hence also in genbank. This is further
corroborated by the fact that also 91% of the reads of a metagenome from the US mapped against our
database (see also Supplementary Fig. 5).

I am a little confused about the definition and analysis of the variome. From what I understand the variome is
composed of the all the genes present at less than 10% frequency of the core genome of each phylotype.
The authors then look at the association of these frequencies to the frequency of the different SDPs. Isn't
there a methodological flaw here? If some SDPs are found at 30% frequency in a phylotype, it is then
impossible for any genes of the variome to be defined as a "core gene" of that SDP, since the genes of the
variome can only reach 10% frequency (by definition). An alternative approach could be to look for the
genes that are present in the same frequency as the genes of the SDPs and compare that to the random
expectation (e.g. relative to random frequencies). This might help the authors identify whether there are
SDP-specific genes.

**Reply:** The variome is composed of all the gene families present at less than 10% of the phylotype
abundance in at least one sample. In other words, a gene family is assigned to the variome if the coverage
is considerably lower than the phylotype terminus coverage.

The rationale is the following: Gene-families carried by all strains of a phylotype (and therefore all SDPs) are
expected to have a coverage equal to (or higher) than the phylotype terminus coverage (i.e. 100%). We use
a conservative threshold to take noise related to sequencing and mapping into account. If the coverage of a
gene-family is less than 10% of the phylotype terminus coverage, it is highly unlikely that the family is carried
by all strains of the phylotype within the sample. Thus, the gene family is "variably associated with the
phylotype".

Because the variome analysis is carried out at the level of the phylotype and not at the level of the SDP,
there is no methodological flaw. For example, if an SDP is present at 9% in one of the samples, then an
SDP-specific core gene family (i.e. a gene family that is present in all strains of that SDP but not in any of
the other SDPs) can be assigned to the variome, as it will have a frequency of 9% relative to the phylotype
abundance.

The reviewer suggests looking for genes that are present in the same frequency as the genes of the SDPs.
We have in fact have carried out such an analysis, by determining the pearson correlation coefficients
between the raw abundance of the variome gene families (log10-transformed) and each SDPs (SDP
terminus coverage, log10-transformed) (see lines 322-327, Figure 6a, Supplementary Data 6).

We have detailed workflow of this analysis in Supplementary Fig. 15, and made several edits in the revised
text, which we hope have clarified our approach.

The term "variome" was new to me, although other terms have been used to define the same notion (e.g.
"the accessory genome "or "the flexible genome"). I don't think we need to introduce a new word that would
add more confusion. Also, when googling "variome" it appears that this term is predominantly used to refer
to the variable SNPs within the human genome, which is quite different from what the authors are talking
about.

**Reply:** We agree with the reviewer that there are alternative, more broadly used, terms for the same notion.
However, as we use the term quite often, also in combination with other nouns, a simple word like 'variome'
makes for easier reading (e.g. "gene families of the flexible genome" vs "variome gene families").
In fact, there are a number of publications where the term "variome" was used in the same context
(<https://doi.org/10.1186/s12864-017-4376-0>; <https://doi.org/10.1093/molbev/mst048>;
<https://doi.org/10.1111/1462-2920.13455>). To avoid any confusion, we define the term "variome" in the
revised manuscript at its first occurrence on line 315-320:

*"Following coverage normalization, gene families occurring with a relative abundance of less than 10%*
*compared to the phylotype abundance in at least one sample were categorized as "variably associated with*
*the phylotype" (see methods and Supplementary Fig. 15). Based on this conservative threshold, a third of*
*the gene families in the current database (5,059; 32%) displayed variable phylotype association (Fig. 6a,*
*Supplementary Data 6). In the following, we will refer to these gene families as the "variome"*

The authors wrote they removed very similar genomes from their databank, but they don't explain what
criterion has been used to define the very similar genomes.

Reply: Very similar genomes were removed by inspecting the genome-wide phylogenies and by looking at
the pairwise ANI values. We also excluded genomes that were of low quality (i.e. too fragmented
assemblies) or originated from single-cell sequencing (risk of contamination and highly fragmented). We
have added the following sentence to the methods section to clarify how the database was streamlined
(L559-561):

*"Based on this collection of genomes, the database was streamlined to reduce redundancy due to highly*
*related strains (maximum 98.5% pairwise ANI), prioritizing the most complete genome assemblies."*

For mapping, genome similarities above this threshold are unlikely to recruit more reads. In this context, it is
also worth noting that most metagenomic studies use a single genome representative per
species/phylotype, whereas our database contains up to 14 genome representatives per phylotype.

Reviewer #2 (Remarks to the Author):

The study by Ellegaard and Engel investigates the intra-population diversity among common microbial
residents of the bee gut microbiota. For this, they use a reference genome collection for major 'phylotypes'
of the bee gut (which includes novel genomes they sequenced from their own lab stock collection) to recruit
reads from honey bee gut shotgun metagenomes, which are generated from individual bees that cover
multiple age groups and colonies. The study is overall well-written and clear, figures and tables are well-
prepared.

While I commend the authors for their excellence in generating such a valuable dataset which brings
insights to many fundamental questions of microbiology within reach, I am left somewhat disappointed with
their analysis of the data, which essentially led to rather uninteresting findings that can be summarized to
this: 'we have culture representatives for the vast majority of the most abundant phylotypes of the bee gut,
yet microbes of some bees are not identical to the cultured isolates we have'. I don't have the impression
from their text that their analysis of these data addressed any of the important questions they postulated in
their introduction.

**Reply:** We thank the reviewer for the complements on the writing and the dataset. We also believe that this
dataset has great potential for a wide range of analysis, and hence are convinced that it can be mined
further to address additional questions about genomic diversity in natural bacterial populations.

As for our results; While we do spend a paragraph demonstrating that the database is exceptionally
representative of the community, this is not actually a main results. Rather, it was done to ensure that the
downstream analysis is sound, and to justify the use of a reference-database guided analysis. The reviewer
concludes that the only finding of our study is that '*we have culture representatives for the vast majority of
the phylotypes, but that microbes of some bees are not identical to the cultured isolates*'. While we
respectfully disagree with the reviewer that this is the only conclusion from our study, it indicates that our
main findings and their relevance were not well communicated. In the following, we provide a summary of
the two main questions we have posed in the introduction of the manuscript, and show how our results
addressed these questions. We then explain how we have revised the manuscript to communicate our
findings and show their relevance. Finally, we list three novel analyses that have been included in this
revision, which provide further detail and insights to our original findings.

Overall objective of our study (as stated in the introduction, L29-30): "*... how diversity is structured.. in*
*natural microbial communities*"

We used the honey bee gut microbiota as a model to address this overall objective. The honey bee gut
microbiota is ideally suited due to its low-complexity at the 'phylotype' level and the high extent of 'strain-
level' diversity (Engel et al PNAS 2012). The community is highly specialized to the bee gut environment
and has coevolved for >80 million years with its host (Kwong et al 2017 Science Adv), providing us with the
possibility to study a community with a defined natural habitat and known evolutionary history. Moreover,
bees are essential pollinators whose gut microbiota has recently been shown to affect bee health in various
ways (Raymann et al 2017 PLoS Biol, Zheng et al 2017, PNAS, Motta et al 2018 PNAS, Kesnerova et al
2017 PLoS Biol). For all these reasons, we believe that investigating this overall objective in the bee gut
microbiota is highly relevant to a broad scientific community.

Specific question 1 (as stated in the introduction, L30-32): "*Are bacteria organized into genetically and*
*ecologically congruent units, akin to the species of the eukaryotic world? And if so, how can we delineate*
*these units? "*

*Our findings:* Focusing on the honey bee, we show that the gut microbiota harbors very high levels of
diversity within phylotypes, which is structured into discrete evolutionary lineages (SDPs). Some SDPs are
exceptionally diverse, and do not conform to previously established boundaries, raising new questions about
diversification/speciation processes. We also show that genes encoding relevant functions in terms of e.g.
carbohydrate metabolism are variably present within phylotypes, but are often not bound to a given SDP.
Moreover, such genes are frequently carried on genomic islands, thus pointing in the direction of horizontal
gene transfer as an underlying mechanism of functional diversification.

These results address the initial question, as we show that phylotypes are structured into genetically
congruent units (the SDPs), which harbor additional diversity at the strain level. With our systematic
approach, we demonstrate how to delineate these units using metagenomic assemblies and a reference

database that is representative of the community. In addition, we also take a stab at the question of whether
these SDPs are ecologically congruent: specifically, we identified a subset of gene families that were both
specific to a given SDP in the database, and correlated in abundance with the SDP across the metagenomic
samples, which are promising candidates for ecological niche differentiation.

Specific question 2 (as stated in the introduction): Do closely related lineages co-exist, and if so, how?

*Our findings:* Having addressed how diversity is structured in the bee gut microbiota (see findings to
question 1), we next addressed question 2, by analyzing how SDPs and strain-level diversity distribute
among individual bees of different ages and colonies, over a time period of two years. We find that most
SDPs co-exist within individual bees (except for the case of one gammaproteobacterium phylotype,
Gilliamella). In contrast, we find segregation into individual bees at the strain-level. We also show that
functional gene content is variably present among bees, as a consequence of the strain-level segregation.
Overall, these findings illustrate, for the first time, that individual honeybees harbor distinct bacterial
communities. While some impact of age was observed, we found that bees with the same age, sampled at
the same time and colony, still carry distinct communities.

These results precisely address the initial question of whether closely related strains co-exist. We agree that
the added question of how this may happen is not resolved by our study. However, our results provide
testable hypotheses (ecological differences at SDP level and priority effects during community assembly),
which can now be addressed experimentally, using the bee model.

*Importance:* we believe that an understanding of the structure and distribution of genomic diversity in natural
bacterial communities is fundamental for understanding both their evolution and function. Given the difficulty
of obtaining such data, there is currently precious little information available from other studies, which is
another reason why these results should be of interest to scientists in other fields too. Moreover, our findings
are highly important for future research on the bee gut microbiota, a community that has received great
attention in recent years. As stated, the bee gut microbiota is experimentally tractable and can therefore be
used to address a broad range of questions about the function, ecology, and evolution of host-associated
communities. Moreover, recent findings about the relevance of the bee gut microbiota for bee health show
the need for further understanding of the structure and distribution of the community, to guide experimental
approaches and test the impact of bacteria on bee health.

*Revision in manuscript:* We have carefully edited the manuscript text, to more explicitly state the findings
and their relevance. Specifically, we have added introductory and concluding sentences to each results
section to explain the scope of the analysis and its relevance. We have made a major revision of the
discussion, focusing more on general concepts of microbial diversity and how our study helps to understand
how diversity is structured. At the same time, we have also tried to make it more apparent how our study
advances the field of bee microbiome research.

*Novel analysis:*

(i) We have included a database-independent analysis that shows increased beta-diversity in winter bees
compared to young and middle-age bees (by clustering universally conserved single copy ortholog gene
sequences, extracted from the metagenomic assemblies). This provides further evidence for the pattern
reported previously, which was based solely on the mapped metagenomic data.

(ii) We have expanded the analysis on the distribution of single nucleotide variants (SNVs) across individual
bees. Specifically, we have calculated the fraction of variable sites found among bees of the same sampling
(same age, colony and collection time), compared to random subsets of bees. We find no significant
differences between real and random samplings, indicating that the observed individualized communities are
not a consequence of sampling variability.

(iii) We have carried out a rarefaction analysis of SNVs. This analysis illustrates that about 20-30 bees are
needed to sample most of the strain diversity (based on SNVs) present in our dataset. By generating
multiple curves per SDP, we show that the sampling order has little influence on the pattern. Moreover,
since the same pattern was seen for all SDPs, the analysis indicates that each bee harbors roughly the
same fraction of the diversity in the colony.

The most plausible explanation for the similarity of rarefaction curves across SDPs is that a common
underlying process determines how much of the total diversity can be present in a single bee. We discuss

possible scenarios that could lead to such a pattern, and propose that neutral processes during the
community assembly phase (i.e. priority effects) could be responsible. Bees are social insects, constantly
engaging in social interactions with each other, which should in theory provide ample opportunities for
exchange of strains. However, it is also known that bees are born germ-free and rapidly acquire the
microbiota over the first few days. In fact, a few hours of exposure to the colony is sufficient to get bees fully
colonized. Hence, if there is a critical time window for colonization, in which bees just probe a fraction of the
available strains of the colony, then initial differences in colonization may not be equalized by later
encounters, and a pattern as observed in our study could emerge. We discuss this and other possibilities in
the revised manuscript and refer to other studies that have recently been published in this context (e.g.
Laventhal et al Nat Microbiol 2018).

We hope that with the manuscript edits, the additional analysis, and the explanations given in this point-by-
point reply, we can convince the reviewer that our findings are interesting and relevant to a broad scientific
community.

The number and average identity of sequence-discrete populations given the reference collections can
indeed tell us about the coverage of our isolates, which in almost every single environment leads to the
same conclusion: metagenomes contain more diversity compared to what we managed to cultivate, and 16S
rRNA gene amplicons are not suitable targets to characterize it. But this strictly reference-based approach
does not tell us anything about 'why' it is the case, which is the part that is not yet established. With its
superb experimental design the study gets very close to taking a stab at fundamental questions that could
be relevant to the broader community of microbiologists, yet it falls short. These may be interesting findings
for the bee community, but in my opinion the scope of findings do not go beyond that in my opinion as I don't
see how recruiting reads from pooled individuals helps us see anything besides SDP curves that have no
biological meaning. I believe this dataset could have revealed much more regarding the gut microbial
evolutionary and ecological dynamics of bees within and across colonies if the authors reconstructed
individual genomes from individual bees and use established practices of comparative genomics, which
could shed light into the accessory genomic regions and how they compare to isolated representatives of
phylotypes and to each other across different individuals.

**Reply:** The reviewer raises three important points in this paragraph, which we will address one by one as
follows:

326 1. Diversity in metagenomes vs in cultured isolates

We agree with the reviewer that a strictly reference-based approach will not tell us anything about why we
see more diversity in metagenomes compared to cultured isolates. This is an interesting question, but it was
not the scope of our study. In fact, the read recruitment to our reference database was exceptionally high
(about 90%) compared to other studies, justifying its use for the questions we wished to address (see
above). Furthermore, we want to emphasize that our approach was not strictly reference-based. We used *de*
*novo* assemblies for several analyses, including the new beta-diversity analysis (line 211-222, Figure 4b)
and the SDP validation.

335 2. SDPs and their biological meaning

The presence of SDPs, with the level of discreteness observed in the current study, translates into separate
evolutionary lineages. In other words, these populations are, at this point, evolving as independent
populations. This is obviously a fundamental piece of the jigsaw, when it comes to understanding the
composition and evolution of the community (which is, in itself, an important objective). However, we also
think it is reasonable to assume that something triggered the divergence in the past, and that there is a
biological reason behind the apparently stable current co-existence of these SDPs, not only in the colony,
but even within individual bees. Our analyses clearly go beyond showing the existence of SDPs: we
analyzed (i) how SDPs distribute in the host population and (ii) which fraction of the flexible gene content
(the 'variome') correlates with SDPs. Both of these analyses provide biologically relevant insights about the
existence of these SDPs. Apart from this, we agree that we cannot unambiguously determine what the
relevance of these SDPs is (e.g. for the bee, or the stability of the community), or how they came into
existence, using our metagenomic approach. However, now that we know that SDPs exist and how they
distribute, we (and others) can start testing possible hypotheses using experiments.

350 351 3. Reconstruction of bacterial genomes from individual bees

The possibility of generating MAGs (metagenome-assembled genomes) represents an exciting new field,
and fulfills an important role in providing genomic information for species, which could not have been

obtained in any other way (as the majority of bacteria in natural communities cannot be cultured). In our
case, however, all of the phylotypes known to colonize the honey bee have been cultured, which in turn has
facilitated the establishment of a highly comprehensive database. Therefore, we are in the privileged
position of being able to choose another strategy.

As stated above, we have not only relied on the reference database. De novo assemblies were in fact done,
for all samples individually, and they fulfill an important function in the current study (validation of SDPs and
beta-diversity analysis). However, when we explored the possibility of binning these assemblies at the
beginning of the study, we encountered several problems, which led us to the conclusion that a database-
driven analysis would provide a more robust analysis.

Metagenomic assembly is well known to be challenging, with the main problems being shallow coverage
and strain-level diversity (Sczyrba et al. 2017, Nature methods). While we do have a high coverage for our
community members, we also have exceptionally high strain-level diversity (perhaps because of it?), even
within individual bees (Figure 5b). For example, the mean fraction of variable sites in the core gene
sequences of Bifido-1 within individual bees is nearly 11% (not including rare SNVs). Thus, while strain-
diversity was found to be lower within bees, the diversity is still very high. This represents a severe problem
for any assembler, and in the following, we present an analysis that illustrates how it impacts our
assemblies.

In the course of our SDP validation analysis, we extracted orthologous sequences (ORFs) of all phylotype
core gene families from each metagenomic assembly (i.e. all samples). Since we included the sample
affiliation in the fasta headers of the ORFs, the number of sequences extracted for each family can be
counted from the headers. The results for the bifidobacteria are shown in Figure 1 in this reply, and is
explained further here below. The results for the other phylotypes are included on Zenodo (SDP
validation/plots; doi: 10.5281/zenodo.1479668).

In theory, all the core gene families should be present in a single copy within each bin in a sample (as they
are in the sequenced isolates), and the expected number of bins should therefore correspond to the number
of core gene sequences per family. With a well-behaved assembly, we should see a single main "blob" per
sample, corresponding to the number of bins present within that sample. In practice, for the Bifidobacteria in
particular, we see multiple "blobs" for all samples, with some samples spanning a wide range (e.g. sample
GrY2_N4, which contains 1-6 orthologs per family). Similarly, we get 4-5 estimated bins for the Firm5
phylotype in the majority of samples, likely associated with the Firm5-4 SDP (for which the mean fraction of
variable sites in the core gene sequences was also high within bees, 7%).

The patterns in Figure1 could be due to highly fragmented assemblies/mis-assemblies (we require 50%
query coverage on extracted orthologs), but they may also be a consequence of having multiple strains with
divergences falling at the borderline of the capacity of the assembler. Specifically, some core genes may be
sufficiently similar to yield a single sequence, while others are sufficiently divergent to generate multiple
sequences, for the same set of strains. Whichever mechanism is at fault, it means that MAGs with a decent
level of completion or correctness will be very challenging to obtain for our data.

For most of the remaining phylotypes, we found that the expected number of bins approximate the number
of SDPs that we have identified. We can therefore also conclude that the strain-level diversity found within
SDPs, which we have quantified and shown to be extensive, typically gets collapsed into single consensus
sequences. Using classical comparative genomics tools on such sequences (i.e. phylogenetic inference, or
selection screens) is therefore problematic. Moreover, considering that most of the accessory gene content
does not appear to be conserved within SDPs, we expect that the majority of these functions will not be
binned at all. In conclusion, while we agree that it would be fantastic if we could pull out near-complete
genomes from within the SDPs, our data strongly indicates that this is not feasible with short-read shotgun
metagenomic data and the amount of strain-level diversity present in each SDP in our samples.

Figure 1: Core gene counts in metagenomic samples. Core gene sequences were recruited based on all genomes of the "Bifido" phylotype in the database. The coverage of Bifido-1 and Bifido-2 is shown in the left panel, since it also has a strong impact on the assemblies. The presence of multiple similar sized "blobs" in the right panel shows that the core gene families generate different numbers of bins within samples.

Nevertheless, we have identified approximately 5000 gene families that belong to the accessory gene pool in the current study (the 'variome'), and made a detailed analysis of their distribution across bees. Moreover, with the aid of sequenced isolates (which are expected to be complete and correctly assembled), we have also shown that most of the variome occur in genomic islands (341 big islands are listed in Supplementary Data 7). This demonstrates the advantage of using a reference database, as we would not have been able to identify genomic islands without complete reference genomes. We therefore think that our study, and especially the variome and genomic island analyses, do shed light into the accessory genomic regions, very similar to what the reviewer proposed.

Another surprise for me was the statement in line 438: "The genomic database, bed-files, gene sequences/identifiers, ortholog predictions, annotations, and key scripts will be made available in a public repository upon publication". By not making available their data and ad hoc scripts to reviewers, perhaps authors are suggesting that they do not trust reviewers' ability to not take their stuff and run away. I am not

sure how reviewers are expected to distinguish fiction from science when data and scripts are not accessible
for inspection. I find this attitude not only insulting, but also against the nature of scientific review. There are
many services that would allow anonymous access to data and scripts without making them public, and "we
will make them available after publication" should not be acceptable for credible journals that strive for
excellence during the review process.

**Reply:** We are very sorry that this issue came up, as it is of utmost importance for us that our work gets
properly reviewed. We simply did not expect that a reviewer would want to spend the time to look into the
raw data and the scripts. But we definitively appreciate this commitment of the reviewer and would have
been happy to provide the scripts/data for the first review. Our intention was really not to hide anything or to
mistrust the reviewer. So, we have gone to great lengths to make everything transparently available and
reproducible for this revision.

First, we have made all the raw data public on the sequence read archive (SRP150166). Second, we have
made the full pipeline available on Zenodo (doi: 10.5281/zenodo.1479668), including all scripts used, and all
final result files. Some intermediate files (e.g. bam-files) have not been uploaded (due to size limitations),
but they can be reproduced from the raw data, using our scripts. All perl scripts come with documentation,
bash scripts have been included to show their usage as part of the pipeline, and README files are included
in all directories. We hope the reviewer will find our documentation satisfactory, and encourage the reviewer
to ask the editor directly if, despite our efforts, any additional information or clarification should be needed.

We thank the reviewer for this comment as we agree that research should be transparent and reproducible.

Besides these major points regarding the focus and the depth of the study, I would like to thank the authors
for their meticulous data analyses. While I believe their strategies led to rather uninteresting findings, high
level of meticulousness for every stage of the analysis is quite impressive and exemplary.

**Reply:** We thank the reviewer for these nice words and hope that with the above clarifications and our effort
to revise the manuscript (edited text and additional analyses), the reviewer will now not only acknowledge
the meticulous analysis and superb quality of the datasets, but also appreciate the relevance of our findings.

Some minor points:

182: Reminiscent of dysbiosis? How do authors define dysbiosis, and based on what they link increased
abundance of microbes that do not match to known phlotypes and decreasing replication rates for the ones
that are known? Aging could be correlated with changes in diet and/or immune system functioning. Why is it
necessary to link it to a buzzword that is not even properly defined and understood where it is studied most
extensively? Overall this section seems disconnected from the title. The finding here is simply that the
culture collections are more representative of younger individuals. I think the authors should consider
defining what they mean by this term clearly and discuss it more carefully.

**Reply:** The criticism of the reviewer is justified. We have revised as follows:

First, the fraction of the bacterial community that does not map to our database is actually similar across age
groups. We realize that this was not very visible in our previous figures, and we have therefore added a new
panel in Supplementary Fig. 2 to show this more clearly. There is a statistically significant increase in the
amount of unmapped reads, but its a small increase, we are still mapping around 90% of the non-host reads
in the old bees. Hence, the culture collection is highly representative of both young and old bees.

Second, the reason why we only display PTR relative to age for the five core phylotypes in Fig. 4 is that they
occur in essentially all bees. In contrast, *Bartonella apis* was only found in two young bees, so an age
comparison was not feasible. Since the five core phylotypes are always there, we find it interesting that they
behave differently in winter bees compared to younger bees. A lower average population replication
suggests that the community is less active and therefore may not be as effective in providing colonization
resistance against opportunistic gut bacteria (i.e. bacteria not belonging to the core phylotypes). We have
now more explicitly indicated this possible link in the results on line 219-222.

We used the word dysbiosis in the meaning of an "un-balanced" community, i.e. more variation in
community composition between bees. For example, *Bartonella apis* reached very high relative abundances
in the winter bees in one colony, but not in the winter bees in the adjacent colony (see Fig.S8, right-most
column). To formally test this, we have included a beta-diversity analysis in the revised manuscript. For this

analysis, we extracted all orthologs of a defined set of 33 universally conserved single-copy genes
(Sunagawa et al. 2013, Nat. Methods) from the *de novo* metagenome assemblies, and clustered them at
95% sequence identity. The prevalence of these clusters across bees of different age was then used as a
measure for beta-diversity, i.e. the difference in community composition between bees. As expected, we find
that winter bees harbor significant higher levels of beta-diversity, meaning their gut microbiota is less
homogeneous than in young bees (Figure 4b). We agree that the term "dysbiosis" has been used
ambiguously in some studies and is not well defined. Therefore, we have substituted it with the term "un-
balanced".

We have rewritten the section about the different evidence pointing towards an un-balanced microbiota in
old winter bees compared to young bees and include the new beta-diversity analysis.

Another such word that makes a surprising appearance is 'speciation'. Speciation appears three times in the
manuscript, one in each of the following sections: the abstract, results, and discussion. The results section is
where the justification is needed the most, yet lacking. The authors associate differences in intra-SDP
diversity among core genes of representative genomes through read recruitment with stages of speciation.
Most microbiologists could agree with that statement, and I would have let it go if I had read it in a Tweet.
But here a stronger justification could be beneficial.

**Reply:** We have revised the result and in particular the discussion substantially for this question and also
replaced the term 'speciation' by 'diversification'. We agree that the meaning of that term remains vague if
not further explained.

To summarize, we identified three SDPs that are exceptionally diverse, also compared to the other SDPs
analyzed in the current study, with which they share their habitat. This increased diversity is supported by
the divergence among the isolates in the database (see long tip branches in Fig. 2) and the recruitment of
assembled core gene sequences to the database (see wide curve in Fig. 3 for Bifido), but also from the
quantification of SNVs within SDPs (see Fig. 5).

We have added the total coverage for each SDP (as quantified on the reduced database) to Figure 5b. This
plot illustrates how deeply the SDPs were sequenced across the study, and therefore how well the SDPs
were sampled overall. Notably, a previous study on the human gut microbiota found that very few new SNVs
were discovered after 1000x coverage (Schloissnig et al 2013, Nature), a threshold that we are well beyond
for most SDPs. Moreover, the plot also shows that the most diverse SDPs are not the most deeply
sequenced ones, indicating that our read coverage normalization strategy worked, and that the differences
are real.

So we have three SDPs that are exceptionally diverse, with ANI values larger than previously reported for
other SDPs. It is always difficult to discuss and present results related to bacterial species, but the patterns
we see are nevertheless what we would expect, if diversification/lineage splitting (we called it 'speciation' in
our previous manuscript) is on-going.

Showing more examples of the underlying coverage data from which PTRs were assessed as
supplementary figures could help. Because the observation of decreased coverage may be a by-product of
decreasing coverage of the populations authors have access to. As authors also mentioned, Figure 4 shows
decrease in coverage for almost all phylotypes and increase in unmapped reads in elder bees. I think it is
important to solidify the correlation between decreasing coverage and PTR to avoid any potentially
misleading claims.

**Reply:** We agree that it is important to take all possible biases into account. The coverage plots of all
samples were provided in Supplementary Data 2 (now in Supplementary Data 4), the coverage is on the y-
axis. Overall, we find that the linear regression fit gets noisy at very low coverages, and it is for this reason
that we did not include samples with less than 10x phylotype coverage in Fig. 4. At coverage >10x, there is
no clear association between coverage and PTR. In fact, if anything, there is a negative correlation (with
higher PTR values for lower coverage), see new Supplementary Fig 7. Which is actually interesting too, but
we would like to have some more data, before we start speculating about this trend. For the scope of the
point raised by the reviewer, we conclude based on this new analysis that there is no positive correlation
between coverage and PTR.

306: I think the argument that there is a universally conserved make up of the bee gut microbiome beyond
geographical distances is conceivable. But isn't it a bit too much to say in this level of conviction when the

only additional data are coming from a single sample that pools a bunch of bees in the US? What is the
genetic makeup of those bees in the US sample? Maybe being more cautionary would better serve
everyone.

**Reply:** We fully agree that a single sample from the US is not sufficient to make strong statements about the
universal distribution of these SDPs. We hence tuned down the language in the results (L173-175) and the
discussion (399-404).

312: I do not agree with the authors' suggestion here. Genome-resolved metagenomics, phylogenomic and
comparative genomic analyses of resulting population genomes would be much more suitable for attempts
to delineate 'species', whatever that means. Many groups are already doing exactly that, demonstrating the
efficacy of having genomes rather than counting single-nucleotide variants using references. What is done in
this study is simply trying to address a very complex problem through crude associations. The approach
used seems suitable to show how irrelevant / relevant isolate genomes are for a given metagenome, and
even then it is not very useful because it is unable to discuss the presence of accessory genes that may be
in the metagenome but not in isolate genomes, making it impossible to discuss whether any of the isolate
genomes in the collection truly represents the functioning of the population to which it matches with
confidence.

**Reply:** We have removed the sentence on line 312, as this is not a main point of the analysis.

As explained, we agree with the reviewer that reference genomes are irrelevant when the read recruitment
from the metagenomes is low and the database is not representative of the community. But again, in our
case we have 90% read recruitment, meaning that we can recruit most of the diversity with the reference
database. Our reply to one of the previous comments of the reviewer, and especially Figure 1 in this
document, shows that MAGs cannot easily be retrieved for genomes contained within SDP. Therefore, the
SNV analysis is the most accurate way to quantify diversity at this level.

We are fully aware that in other systems, such as in marine metagenomes, this may be a completely
different story and reference genomes are probably much less relevant for metagenomic analysis due to low
representation of the community.

431: It may be the case for the bee community, but other groups that study marine and terrestrial
ecosystems may find these techniques rather coarse. Perhaps there is no need for such qualitative
statements since our level of resolution seems to be increasing rapidly in all fronts.

**Reply:** We have modified the last sentence, focusing on the bee gut microbiota and the general result of our
study:

*"In conclusion, aside from providing fundamental insights into the genomic structure of a host-associated*
*microbial community, our study provides a framework for future studies to test the link between bee health,*
*ecology and microbiota composition at unprecedented resolution."*(L492-494).

Meren.

Reviewers' Comments:

Reviewer #1:

Remarks to the Author:

The authors have adequately addressed all my concerns.

Reviewer #2:

Remarks to the Author:

I would like to thank the authors for their detailed response and excellent revision with additional analyses.

Most critically, I had the impression that the original submission had not addressed important questions postulated in its introduction, and was lacking broader relevance. However, this concern is fully addressed thanks to the additional introductory and summary statements in the Results section, as well as the new points added in the Discussion section. In my opinion these changes made this work even a more worthy read for a broader range of scientists.

Reading the revision and the author response, I also have to admit that one of my suggestions was in fact not quite accurate. I had suggested that a genome-resolved strategy would have been more appropriate, but I think I have missed to what extent the available isolates were able to explain the diversity in shotgun metagenomes. Especially when the reach of the isolates is extended with additional strategies to characterize the within-population genomic heterogeneity and the 'variome', the need for MAGs indeed reduce dramatically. I am not sure if the match between isolate genomes and metagenomes was clear enough in the original submission, by I apologize for missing it regardless. Nevertheless, the revision certainly sets the stage much more quickly to get this message across.

I also thank the authors for making all the raw and intermediate data available along with their ad hoc scripts and for their clarification. I often invest time like many others to have a quick look at the data and scripts on behalf of the community to better point out potential pitfalls. Although, this particular time I am traveling, and I will not be able to go through the data unless I delay my review.

I again thank the authors again for their diligence.

Point-by-point reply

REVIEWERS' COMMENTS:

Reviewer #1 (Remarks to the Author):

The authors have adequately addressed all my concerns.

Reply: Nothing to reply.

Reviewer #2 (Remarks to the Author):

I would like to thank the authors for their detailed response and excellent revision with additional analyses.

Most critically, I had the impression that the original submission had not addressed important questions postulated in its introduction, and was lacking broader relevance. However, this concern is fully addressed thanks to the additional introductory and summary statements in the Results section, as well as the new points added in the Discussion section. In my opinion these changes made this work even a more worthy read for a broader range of scientists.

Reading the revision and the author response, I also have to admit that one of my suggestions was in fact not quite accurate. I had suggested that a genome-resolved strategy would have been more appropriate, but I think I have missed to what extent the available isolates were able to explain the diversity in shotgun metagenomes. Especially when the reach of the isolates is extended with additional strategies to characterize the within-population genomic heterogeneity and the 'variome', the need for MAGs indeed reduce dramatically. I am not sure if the match between isolate genomes and metagenomes was clear enough in the original submission, by I apologize for missing it regardless. Nevertheless, the revision certainly sets the stage much more quickly to get this message across.

I also thank the authors for making all the raw and intermediate data available along with their ad hoc scripts and for their clarification. I often invest time like many others to have a quick look at the data and scripts on behalf of the community to better point out potential pitfalls. Although, this particular time I am traveling, and I will not be able to go through the data unless I delay my review.

I again thank the authors again for their diligence.